# Therapeutic mitigation of measles-like immune amnesia and exacerbated disease after prior respiratory virus infections in ferrets

Robert M. Cox [1], Josef D. Wolf [1], Nicole A. Lieberman [2], Carolin M. Lieber[1], Hae-Ji Kang[1], Zachary M. Sticher [3], Jeong-Joong Yoon [1], Meghan K. Andrews [3], Mugunthan Govindarajan[3], Rebecca E. Krueger [3], Elizabeth B. Sobolik [2], Michael G. Natchus[3], Andrew T. Gewirtz[1], Rik L. deSwart [4], Alexander A. Kolykhalov[3], Khan Hekmatyar[5], Kaori Sakamoto[6], Alexander L. Greninger [2] & Richard K. Plemper [1] ✉

Measles cases have surged pre-COVID-19 and the pandemic has aggravated the problem. Most measles-associated morbidity and mortality arises from destruction of pre-existing immune memory by measles virus (MeV), a paramyxovirus of the morbillivirus genus. Therapeutic measles vaccination lacks efficacy, but little is known about preserving immune memory through antivirals and the effect of respiratory disease history on measles severity. We use a canine distemper virus (CDV)-ferret model as surrogate for measles and employ an orally efficacious paramyxovirus polymerase inhibitor to address these questions. A receptor tropism-intact recombinant CDV with low lethality reveals an 8-day advantage of antiviral treatment versus therapeutic vaccination in maintaining immune memory. Infection of female ferrets with influenza A virus (IAV) A/CA/07/2009 (H1N1) or respiratory syncytial virus (RSV) four weeks pre-CDV causes fatal hemorrhagic pneumonia with lung onslaught by commensal bacteria. RNAseq identifies CDV-induced overexpression of trefoil factor (TFF) peptides in the respiratory tract, which is absent in animals pre-infected with IAV. Severe outcomes of consecutive IAV/CDV infections are mitigated by oral antivirals even when initiated late. These findings validate the morbillivirus immune amnesia hypothesis, define measles treatment paradigms, and identify priming of the TFF axis through prior respiratory infections as risk factor for exacerbated morbillivirus disease.

After years of the COVID-19 pandemic, over 40 million children worldwide are at risk of measles due to delayed vaccination[1] and temporary SARS-CoV-2 viral dominance[2]. Acute measles has a case-fatality rate of ~1%, but lasting immunosuppression constitutes a major health threat after recovery from the primary infection[3,4]. Morbilliviruses such as MeV and CDV invade their hosts through infection of alveolar macrophages (AMs) and dendritic cells (DCs), using signaling lymphocyte activation molecule (CD150) as receptor[5]. Subsequently,

peripheral blood mononuclear cell (PBMC)-supported cell-associated viremia ensues, until viral re-entry into the respiratory tract and infection of epithelial cells through the basolaterally-expressed Nectin-4 receptor[6]. Viral replication during the viremic phase in CD150[+] lymphocytes in peripheral blood and primary, secondary, and tertiary lymphoid tissues depletes CD150-positive lymphocytes, including memory T and B cells, reducing host antibody repertoire and erasing immune memory[3,4]. Such MeV-induced immune amnesia is hypothesized to increase vulnerability to opportunistic infections, leading to significantly increased morbidity and mortality rates from unrelated infectious diseases in the years following primary measles[7]. Treatment options of acute measles are limited to supportive care, IgG therapy in some high-income countries[8], and quarantine of the patient. The exceptionally high infectivity of MeV[9] and vaccine hesitancy resulted in a measles resurgence in many European countries in 2019 even prior to the COVID-19 pandemic[10]. Yet temporary pausing of vaccination campaigns in low and middle-income countries during the pandemic has exacerbated the MeV resurgence and major measles outbreaks are anticipated globally[10]. Despite the severity of this health threat, experimental insight is lacking into the time window for prevention of MeV-induced immune-suppression through direct-acting antivirals (DAAs) and the consequences of a large measles wave coinciding with high activity of unrelated respiratory viruses such as influenza virus and RSV.

We previously developed an orally efficacious morbillivirus polymerase inhibitor, ERDRP-0519, which fully protects ferrets against a lethal CDV infection in a post-exposure prophylactic dosing regimen[11]. We recently identified a structurally and mechanistically distinct broadened-spectrum paramyxovirus polymerase inhibitor, GHP-88309, which is orally efficacious against parainfluenzaviruses and blocks morbilliviruses with sub-micromolar potency in cell culture[12]. Using these compounds, we established in this work measles treatment paradigms and explored the effect of prior disease history on severity of morbillivirus infection.

## Results

To verify efficient oral delivery of GHP-88309 to ferrets, we determined single-dose pharmacokinetic (PK) profiles after oral administration at 50 and 150 mg/kg bodyweight. GHP-88309 demonstrated dose-dependent plasma exposure of 177.8 h × nmol/ml and 754.1 h × nmol/ml, respectively, and sustained tissue distribution exceeding 1 nmol/g tissue 12 h after administration (Supplementary Fig. S1a, b; Supplementary Table S1). Twice daily (b.i.d.) oral dosing at 15 and 50 mg/kg in a 14-day non-formal tolerability study revealed no signs of drug-induced toxicity or abnormalities in serum chemistry (Supplementary Fig. S2a–d) and an abbreviated repeated-dose PK study confirmed consistent exposure on day 7 (Supplementary Fig. S3; Supplementary Table S2). Average trough GHP-88309 plasma concentration in the 50 mg/kg b.i.d. group was 2.4 μM, which was equivalent to >2 × $EC_{90}$ against CDV[12]. This dose of GHP-88309 was used for all subsequent experiments.

### Oral GHP-88309 expands to 8-days the time window of full protection compared to near-exposure vaccination

Efficacy of GHP-88309, ERDRP-0519, and therapeutic vaccination was compared in CDV-naïve ferrets infected with reporter-free, highly pathogenic recombinant recCDV-5804p (recCDV)[13], followed by oral treatment with GHP-88309 (50 mg/kg, b.i.d.) initiated 3, 5, or 7 days post infection (dpi) or oral ERDRP-0519 (50 mg/kg, b.i.d) started 3 dpi (Fig. 1a). Animals in vaccination groups received Purevax CDV vaccine 28, 3, or 1 day prior to, or 1 day after, infection in a prime-boost (28-day group; -28-day prime, -14-day boost) or prime only (all other groups) regimen. A reference group received Distemink CDV vaccine at 2 months of age. There was no statistically significant differences CDV nAb titers between the Purevax prime-boost and Distemink

vaccination groups (Supplementary Fig. S4a). All animals were furthermore fully vaccinated against rabies virus (RABV) by their supplier and confirmed to have α-RABV antibodies prior to use (Supplementary Fig. S4b). Ferrets were monitored daily for clinical signs of morbillivirus disease (fever, rash, loss of bodyweight) and in regular intervals for PBMC-associated viremia, complete blood counts (CBC), and titer of α-RABV neutralizing antibodies (nAbs) (Fig. 1a). All vehicle-treated animals succumbed within 12 days of infection (Fig. 1b) and developed severe clinical signs (Supplementary Fig. S5a–e), whereas GHP-88309 mediated complete survival and statistically significantly reduced virus load (Fig. 1c) when treatment was initiated 3 or 5 dpi, at the onset of viremia and rash, respectively. Treatment started 7 dpi did not significantly improve outcome, and ERDRP-0519 treatment started 3 dpi only partially protected against infection with this wild type recCDV, which is more virulent than the fluorescent recCDV reporter strain used previously[11]. Fully prime-boost vaccinated animals were protected, but neither post-exposure nor 1-day pre-exposure vaccination was efficacious and, moreover, 2 of 3 animals vaccinated prophylactically 3 days prior to infection also succumbed. Clinical signs in animals of all GHP-88309 treatment groups, except the 7-dpi arm, were unremarkable 12 dpi (Fig. 1d; Supplementary Fig. S5), whereas ERDRP-0519-treated animals showed moderate, and vehicle-treated animals severe, clinical signs.

Lymphocyte counts in the vehicle group dropped rapidly by approximately two orders of magnitude by 12 dpi (Fig. 1e). Both GHP-88309 and ERDRP-0519 treatment initiated up to 3 dpi prevented lymphocytopenia. GHP-88309 started 5 dpi did not alleviate initial PBMC collapse, but resulted in populations rapidly recovering such that lymphocytopenia resolved within 4 weeks. Fully vaccinated animals did not develop lymphocytopenia. However, the single surviving animal of the -3 dpi prophylactic vaccination group also experienced temporary collapse of the PBMC population. Pre-existing unrelated humoral immunity, assessed using α-RABV nAbs as a biomarker, was fully preserved in animals of the 5 dpi and earlier treatment groups, but was permanently destroyed in the surviving animal of the -3 dpi vaccination group (Fig. 1f). Thus, GHP-88309 prevents acute lethal morbillivirus infection and mitigates lymphopenia up to 5 days post-infection thereby providing extending the intervention opportunity by at least 8 days compared to near-exposure vaccination. Based on superior oral efficacy of GHP-88309 compared to ERDRP-0519, we selected GHP-88309 for all subsequent experiments.

### Confirmation of immune amnesia hypothesis via recCDV with tunable polymerase processivity

The 100% lethality of recCDV in untreated ferrets within 2 weeks of exposure precluded use of this model in exploring questions relevant to human morbillivirus-induced disease in which lethality is slower and less uniform. To surmount this hurdle, we engineered an attenuated recCDV with intact viral receptor tropism. Previously, we demonstrated that variable length deletions of 39-55 amino acids in the structurally disordered CDV nucleocapsid (N) protein tail domain affect polymerase processivity to different degrees, resulting in genetically stable tunable recCDV attenuation[14]. Employing a 55-amino acid deletion recCDV NΔ425-479 strain, we compared virulence and severity of immune suppression with that of unmodified recCDV and a previously described tropism-altered recCDV Nectin-4-blind mutant strain[3] that cannot engage the Nectin-4 epithelial cell receptor (Fig. 2a).

Clinical signs caused by the different recombinant strains, which were all in an otherwise identical recCDV-5804p genetic background, varied from severe for recCDV NΔ425-479, similar to the parental recCDV, to unremarkable for recCDV Nectin-4-blind (Fig. 2b, c; Supplementary Fig. S6a–d). All recCDV Nectin-4-blind and a majority of recCDV NΔ425-479-infected animals survived, whereas ferrets inoculated with unmodified recCDV died within 12 days of infection (Fig. 2d). Surviving animals mounted a robust α-CDV nAb response within three

 

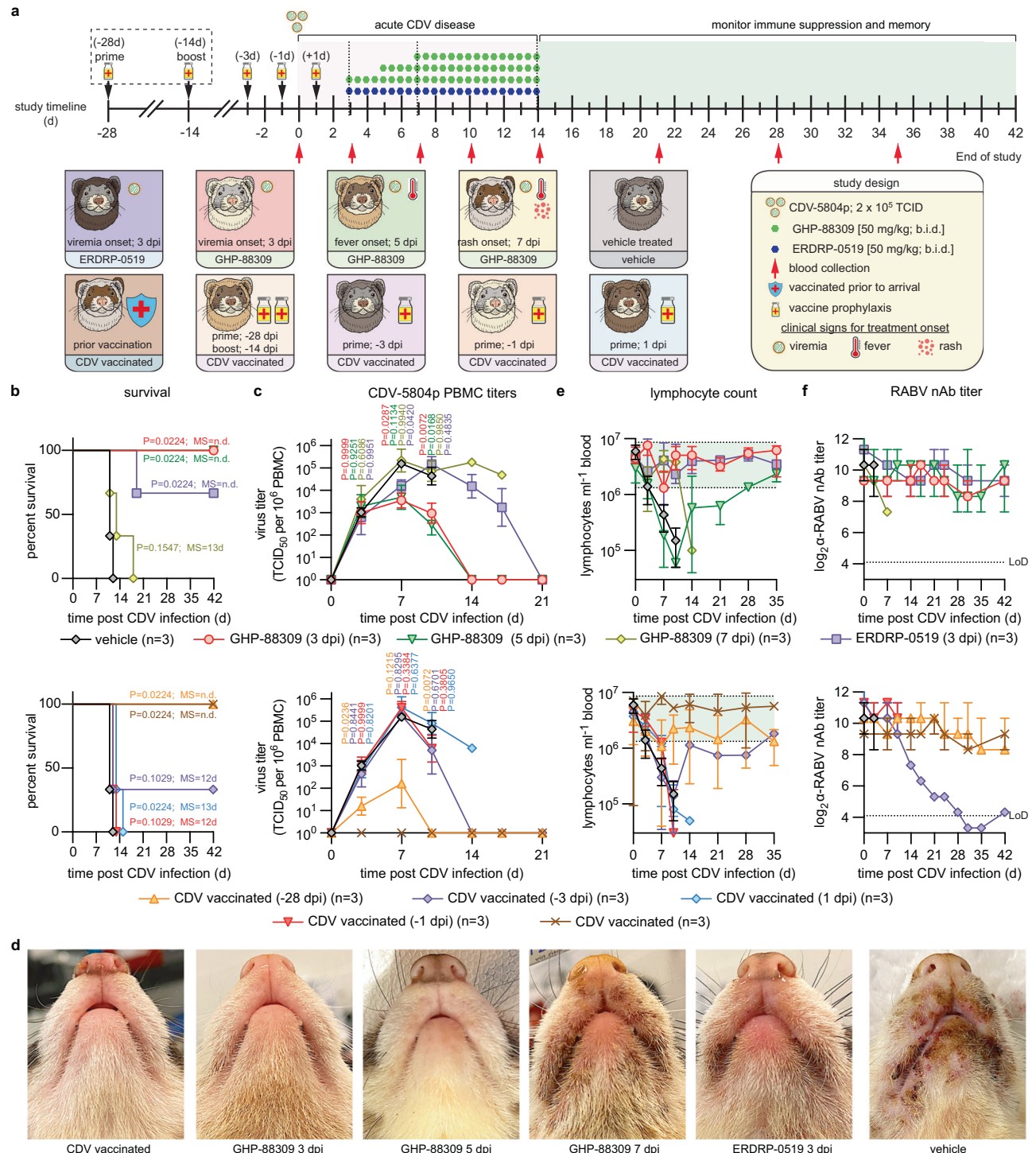

**Fig. 1 | Therapeutic treatment of lethal CDV infection in ferrets.** Ferrets were infected with a lethal challenge of CDV and treated with GHP-88309, ERDRP-0519, or therapeutic vaccination. **a** Schematic of the study design. Ferrets were infected with wild-type recCDV-5804p and monitored for 6 weeks. Symbols in the different boxes show clinical presentation of animals when treatment of the respective group was initiated. **b** Survival curves of ferrets infected in (**a**). log-rank (Mantel-Cox) test, median survival is stated. **c** PBMC associated viremia titers of CDV infected ferrets shown in (**a**). 2-way ANOVA with Dunnett's post-hoc test. **d** Images of ferrets taken 12 days after infection with recCDV-5804p. **e** Lymphocyte counts from infected ferrets measured during the duration of the study detailed in (**a**). Green shading denotes normal range. **f** RABV neutralizing antibody titers from CDV infected ferrets. Symbols (**c**, **e**, **f**) represent geometric means ± geometric SD, lines intersect means. In (**b–e**), top row shows results for inhibitor-treated groups, bottom row for vaccinated animals; LoD, limit of detection; *n* = 3. Source data are provided as a Source Data file.

weeks of infection (Supplementary Fig. S6e). PBMC-associated primary viremia titers of all three strains were statistically identical in the first ~2 weeks after infection until animals inoculated with parental recCDV had succumbed (Fig. 2e). Subsequently, viremia in recCDV Nectin-4-

blind animals fully resolved, whereas ferrets infected with recCDV NΔ425-479 entered a 3-week low-level secondary viremia phase. Only animals inoculated with recCDV NΔ425-479 experienced lymphocy-topenia with initial kinetics similar to that of the recCDV-infected

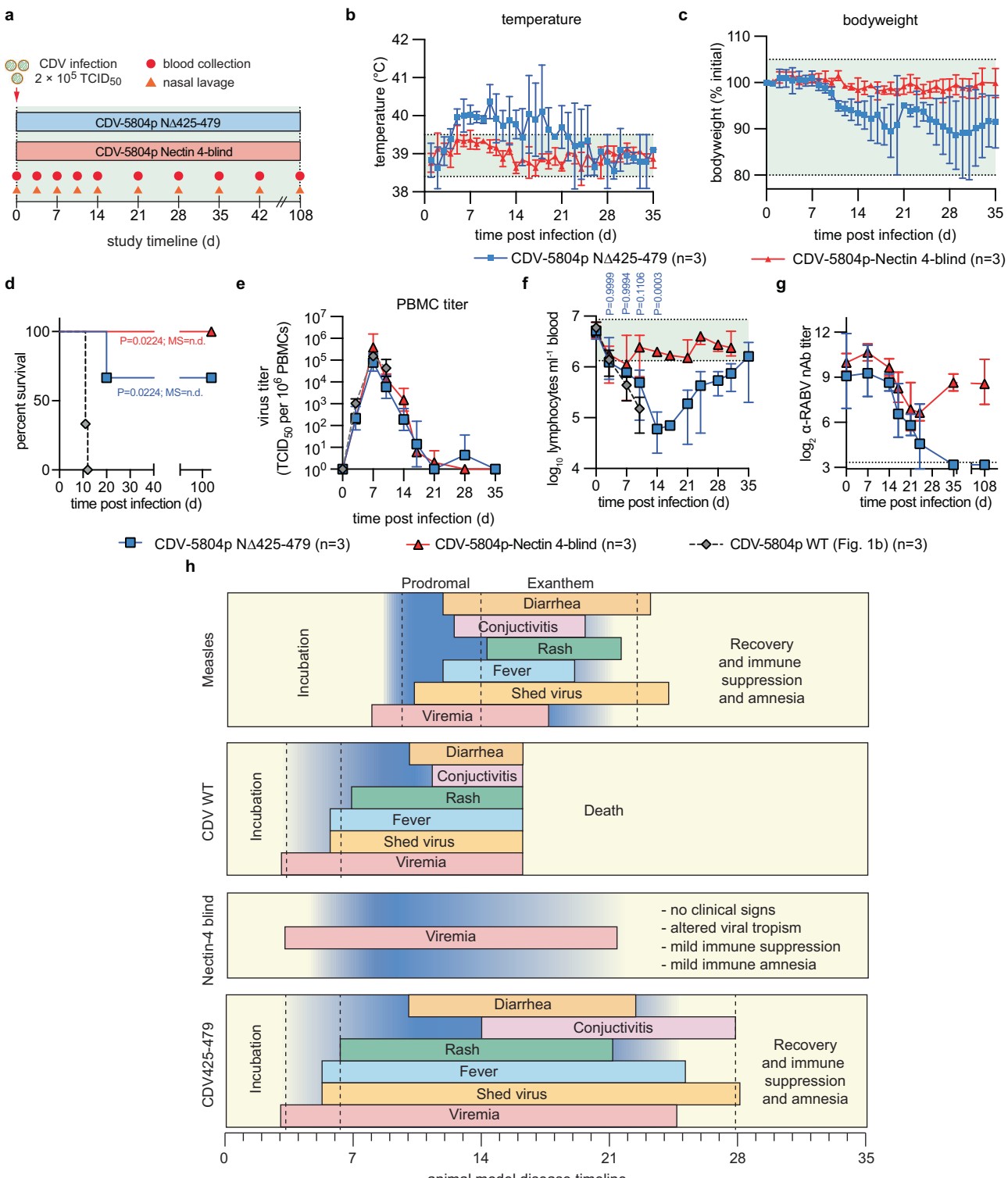

**Fig. 2 | Ferret model of CDV-induced immune amnesia. a** Study schematic. Ferrets were infected intranasally with recCDV-5804p NΔ425-479 or recCDV-5804p-Nectin 4-blind and monitored for 108 days. Nasal lavages were taken from recCDV-5804p NΔ425-479 infected ferrets only. Blood was sampled regularly from all animals. **b**–**g** Body temperature (**b**), bodyweight (**c**), survival curves (**d**), PBMC-associated viremia titers (**e**), lymphocyte counts (**f**), and RABV neutralizing antibody titers (**g**) of ferrets infected with the different recCDV-5804p. Symbols represent arithmetic (**b**, **c**) or geometric (**e**–**g**) means ± arithmetic or geometric SD, lines intersect means; log-rank (Mantel-Cox) test, median survival is stated (**d**); 2-way ANOVA with Sidak's post-hoc test (**f**); n numbers as specified. h, 2D schematic comparing disease dynamics and clinical signs of measles virus infections in humans, with wild-type recCDV-5804p, recCDV-5804p-Nectin 4-blind, and recCDV-5804p NΔ425-479 infections in ferrets. Source data are provided as a Source Data file.

group (Fig. 2f; Supplementary Fig. S7a–d), whereas no statistically significant changes in lymphocyte counts were detected in ferrets infected with recCDV Nectin-4-blind, confirming over-attenuation of this recombinant. In the recCDV NΔ425-479-infected animals, lymphocytopenia fully resolved 4 weeks after infection. Pre-existing humoral α-RABV immunity collapsed within 3 weeks of recCDV NΔ425-479 infection and was not restored over a 3-month post-recovery period (Fig. 2g), whereas recCDV Nectin-4-blind caused only a temporary moderate decline in α-RABV nAbs titers that spontaneously resolved within 4 weeks of infection.

A comparison of clinical signs after infection of ferrets with the three recCDV strains with the presentation of human measles revealed that recCDV NΔ425-479 disease recapitulates hallmarks of human morbillivirus disease, albeit with accelerated onset of clinical signs (Fig. 2h). Thus, we have developed a relevant surrogate animal model of human measles that strongly supports the morbillivirus immune amnesia hypothesis.

## Treatment at the onset of clinical signs alleviates morbillivirus immune amnesia

To explore the degree to which primary clinical signs of morbillivirus disease and immune amnesia can be mitigated pharmacologically, we initiated oral treatment with GHP-88309 at four discrete disease stages after infection with recCDV NΔ425-479: first onset of fever (5 dpi), rash (7 dpi), severe disease with diarrhea and conjunctivitis (10 dpi), and the first fatality in the vehicle group (24 dpi) (Fig. 3a). Treatment was continued b.i.d. until α-CDV nAbs became detectable in serum samples. In addition to following α-RABV immunity, we administered quadrivalent influenza vaccine to all animals 28 days prior to infection with CDV (-28-day prime, -14-day boost). Animals were monitored for 5 months post-CDV and the response to broad innate immune stimulation through intramuscular (i.m.) flagellin, infection with influenza virus A/CA/07/2009 (H1N1) (pdmCA09) and A/WI/67/2005 (H3N2), and homotypic rechallenge with pathogenic recCDV-5804p determined 8, 9.5, 12.5, and 18.5 weeks, respectively, after the original CDV infection.

All GHP-88309-treated animals and all ferrets of a CDV-vaccinated reference group survived until study end, whereas approximately 50% of ferrets in the vehicle group succumbed to recCDV NΔ425-479 within 26 days of infection (Fig. 3b; Supplementary Fig. S8a, b). Oral GHP-88309 started 5 dpi significantly reduced CDV viremia titers (Fig. 3c), but later initiation of treatment did not affect severity or duration of primary viremia. However, first administration of GHP-88309 up to 10 dpi statistically significantly shortened virus shedding into nasal lavages (Fig. 3d). Treatment started at the end of acute disease (24 dpi) did not modulate height or duration of viremia or virus shedding. Lymphocytopenia was suppressed in ferrets of the 5 dpi GHP-88309 group, resembling CDV-vaccine-mediated protection, and alleviated in the 7 dpi GHP-88309 group (Fig. 3e). In animals first treated later than 7 dpi, lymphocytopenia was severe, closely resembling that in the vehicle group, and lymphocyte counts regained pre-infection levels only -10 weeks after CDV infection. Studies in nonhuman primates infected with MeV have revealed that viral RNA remains detectable in PBMCs up to 90 days after infection[15]. Recapitulating this phenotype, we detected CDV RNA for over 64 days in PBMCs extracted from ferrets of all groups, except for CDV-vaccinated animals and ferrets first treated with GHP-88309 5 or 7 dpi (Fig. 3f). Early treatment onset thus accelerated viral clearance and prevented persistence of viral RNA, which correlated with statistically significantly earlier appearance of α-CDV nAbs (Fig. 3g). Whole genome sequencing of viruses recovered from GHP-88309-experienced animals 8-37 dpi with CDV did not reveal emergence of GHP-88309-characteristic[12] resistance mutations (Supplementary Dataset S1).

Upon stimulation of animals with 50 µg flagellin i.m. post-recovery from acute CDV, we monitored expression of a panel of pro-inflammatory markers by PBMCs harvested before, or 1, 2, and 24 h after, stimulation (Supplementary Fig. S9). Body temperature was increased in all animals after flagellin injection without significant differences in marker expression between groups (Fig. 3h), indicating that innate response to an immunogen was not compromised after recovery from acute CDV disease.

Animals of all groups had robust α-RABV (Fig. Fig. 3i) and α-IAV (Fig. 3j) humoral immunity prior to CDV infection. Treatment with GHP-88309 5 dpi fully preserved nAbs titers, identical to the protective effect of CDV vaccination. Animals in all later treatment groups experienced a rapid decline in α-RABV and α-IAV nAbs titers over a 30–40 day period post-CDV infection, which was indistinguishable from that seen in the vehicle group and, in the case of α-RABV immunity, irreversible. However, low levels of α-IAV nAbs were present in the GHP-88309 7- and 10-dpi treatment groups at the time of challenge with pdmCA09 9.5 weeks after CDV infection. Accordingly, animals in both the 5- and 7-dpi GHP-88309 groups showed significantly reduced peak shed influenza virus load (Fig. 3k) and alleviated clinical signs of pdmCA09 infection (Supplementary Fig. S10a, b). All animals infected with pdmCA09 mounted a robust neutralizing response 2 weeks after challenge (Supplementary Fig. S10c), indicating that CDV infection erased pre-existing immunity, but had no lasting suppressive effect on humoral immune competence to a new challenge post-recovery.

Since α-pdmCA09 nAbs are not cross-protective against subtype H3N2 IAVs, they can only provide protection by heterosubtypic immunity[16]. We therefore inoculated animals with an A/WI/67/2005 (H3N2) 3 weeks after the pdmCA09 challenge (87 dpi with CDV) to assess cell-mediated immune competence. As expected, ferrets of the CDV vaccine and 5-dpi GHP-88309 treatment groups had retained high α-H3N2 nAbs titers induced by the quadrivalent influenza vaccine (Supplementary Fig. S10d). Humoral α-H3N2 immunity in all other groups collapsed. None of the groups showed significant clinical signs (Supplementary Fig. S10e, f), α-H3N2 nAbs were rapidly rebuilt within 28 days of A/WI/67/2005 (H3N2) challenge (Supplementary Fig. S10g), and only low levels of virus shedding were detectable 24 h after infection (Supplementary Fig. S10h), suggesting equivalent cross-reactive cell-mediated immunity derived from the pdmCA09 challenge in animals of all groups, which is consistent with previous reports for both the ferret model and human patients[17–21]. All animals furthermore mounted a robust humoral α-CDV response post-recovery and were fully protected against homotypic challenge with non-attenuated recCDV-5804p 130 days after the original CDV infection (Supplementary Fig. S11).

These results indicate that treatment initiated at the onset of first clinical signs of morbillivirus disease (fever; 5 dpi in the CDV ferret model) fully protects, while treatment started at the onset of rash (7 dpi) partially preserves, pre-existing immunity. Later onset of treatment improves outcome of morbillivirus disease but does not subvert immune amnesia.

## Fatal lung disease after unrelated respiratory virus infection followed by morbillivirus invasion

Naturally acquired IAV immunity is more robust than vaccine-induced protection[22]. To better mimic such immunity, we developed a consecutive infection ferret model that establishes a prior disease history in the animals (Fig. 4a). Influenza virus-naïve ferrets were inoculated 28 days before CDV infection with pdmCA09, with the intent to be followed for 5 months including challenge with pdmCA09 and CDV. Animals in reference groups were CDV vaccinated, or not infected with pdmCA09, respectively. Ferrets in all influenza virus groups developed fulminant clinical signs of IAV infection (Supplementary Fig. S12) and reached peak shed viral loads of $10^5$–$10^6$ TCID$_{50}$ units/ml nasal lavage 2 dpi (Fig. 4b). pdmCA09 shedding ceased and all clinical signs resolved by 6 dpi, and ferrets had mounted robust humoral α-IAV (H1N1)

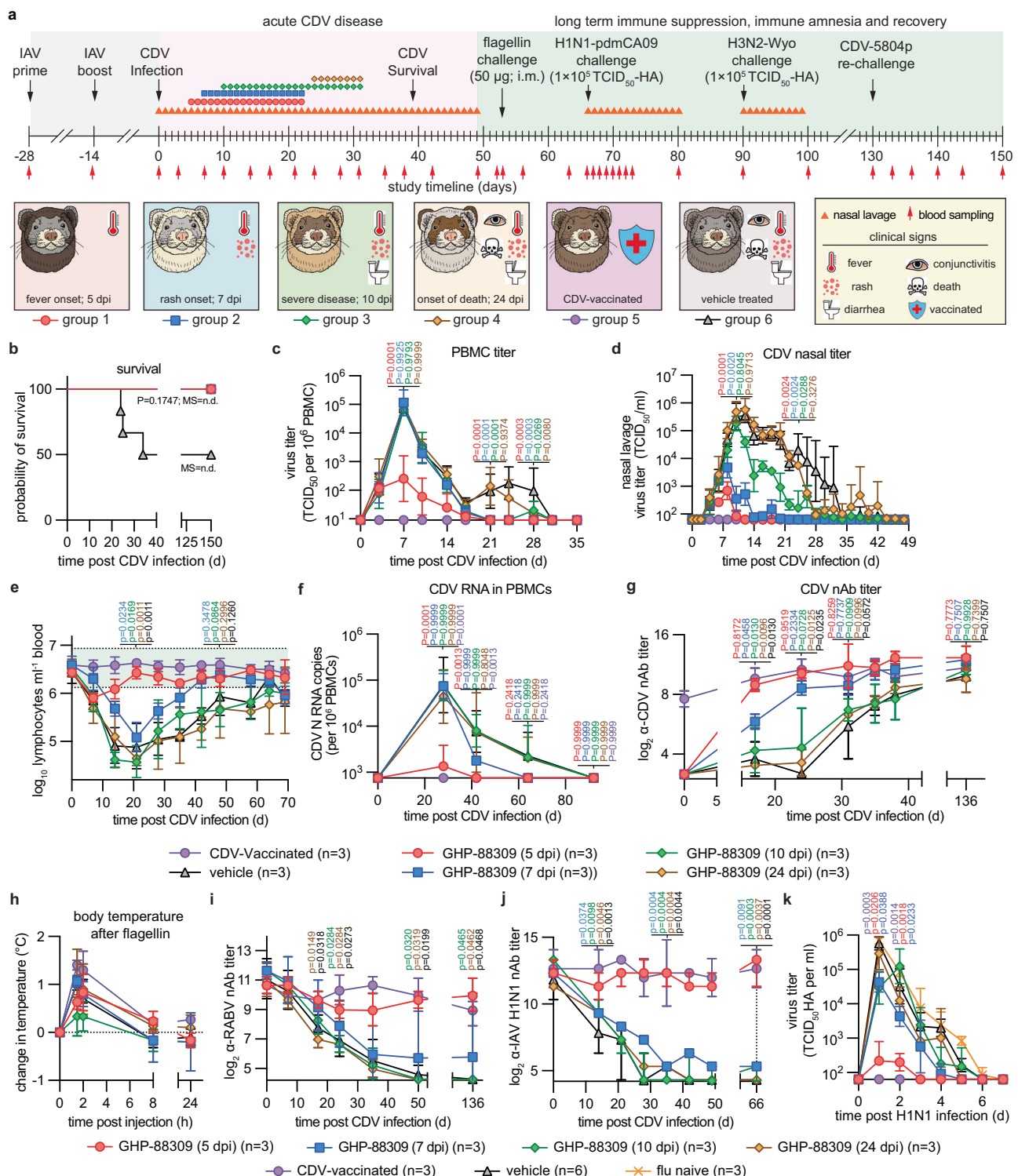

**Fig. 3 | Therapeutic efficacy of GHP-88309 against recCDV-5804p NΔ425-479.**
**a** Schematic of therapeutic efficacy studies utilizing recCDV-5804p NΔ425-479.
Symbols in the timeline refer to the different treatment groups or show nasal lavage
sampling, respectively. **b**–**k** Survival curves (**b**), PBMC-associated viremia titers
monitored until undetectable (**c**), CDV nasal lavage titers (**d**), lymphocyte counts
(**e**), CDV-specific N RNA in circulating PBMCs of ferrets monitored until undetect-
able (**f**), CDV nAb titers (**g**), body temperature changes (**h**), RABV (**i**) and pdmCA09
(**j**) nAb titers, and pdmCA09 nasal lavage titers (**k**) after challenge of ferrets from (**a**)

with pdmCA09 following recovery from CDV disease. Symbols represent geometric
(**d**–**g**, **i**–**k**) or arithmetic (**h**) means ± geometric or arithmetic SD, lines intersect
means; log-rank (Mantel-Cox) test, median survival is stated (**b**); mixed-effect
analysis with Dunnett's post-hoc test (**c**–**e**, **g**, **i**, **j**); 2-way ANOVA with Dunnett's post-
hoc test (**f**, **h**, **k**); multiplicity adjusted P values are shown; green shading in (**e**)
denotes normal range; n numbers as specified. Source data are provided as a
Source Data file.

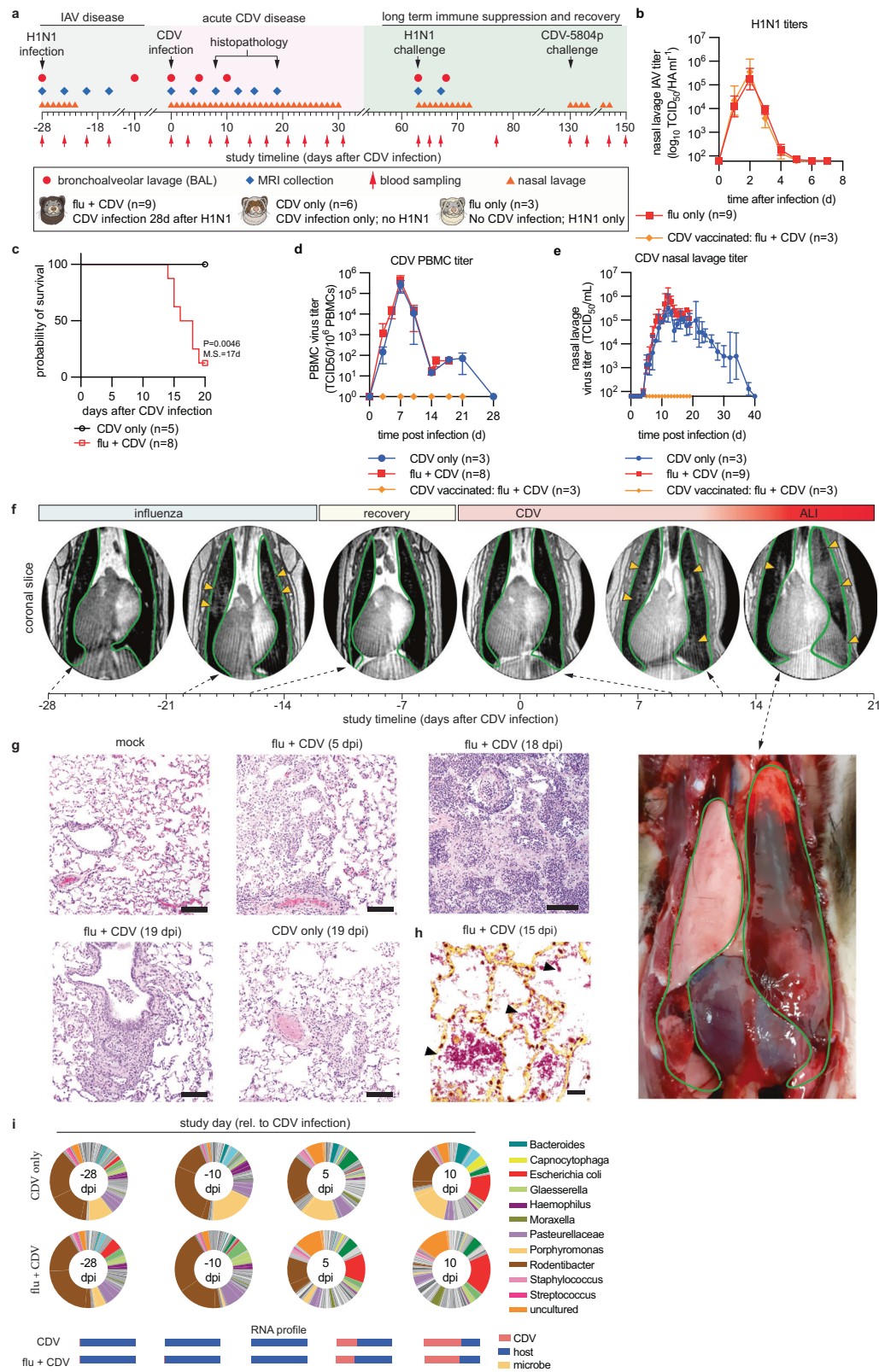

immunity (Supplementary Fig. S13) when infected with recCDV NΔ425-479 4 weeks later (study day 0).

Unexpectedly, most animals consecutively infected with this non-lethal IAV and low-lethal recCDV succumbed to acute hemorrhagic pneumonia with extensive spontaneous bleeding 14–18 dpi (Fig. 4c; Supplementary Fig. S14a–c). Fatalities did not coincide with enhanced PBMC-associated CDV viremia titers or prolonged CDV shedding

(Fig. 4d, e). Postmortems of moribund vehicle-treated animals that presented with severe respiratory distress revealed major lung tissue injury with inflammatory lesions and wide-spread edema affecting several lobes (Supplementary Fig. S14d–f).

We employed pulmonary ferret MRI for longitudinal non-invasive assessment of lung disease progression (Supplementary Movie S1, S2). Transient viral pneumonia with localized fluid accumulation was

**Fig. 4 | Fatal lung disease in ferrets infected with respiratory viruses prior to CDV infection. a** Study schematic. **b** pdmCA09 (H1N1) nasal lavage virus titers after primary IAV infection, 28 days prior to recCDV-5804p NΔ425-479 as specified in (**a**). **c** Survival curves of ferrets from (**a**). log-rank (Mantel-Cox) test, median survival is stated; *n* numbers as specified. **d, e** PBMC-associated CDV viremia (**d**) and nasal lavage (**e**) titers after infection of ferrets with recCDV-5804p NΔ425-479. Symbols in (**b, d, e**) represent geometric means ± geometric SD; *n* = 3–9 as specified. **f** MRI timeline performed on ferrets during primary pdmCA09 (0, 4, 8, and 12 dpi with influenza) and subsequent recCDV-5804p NΔ425-479 infection (8, 12, and 15 dpi with CDV). Lungs are outlined in green (dark appearance in MRI images),

arrowheads denote fluid accumulation. Bottom right: necropsy of the ferret MRI-imaged on D15. **g** Histopathology of ferret lungs from (**a**), extracted 5, 18, or 19 dpi with CDV. **h** Gram staining of lung samples taken 15 dpi with CDV showing bacterial pneumonia. Purple-brown staining indicates Gram-positive bacteria, pink-red color marks gram-negative organisms; scale bars in (**g, h**) represent 100 μm.
**i** Metagenomics analysis of bacterial transcripts in BAL fluids collected from consecutively infected or CDV-only ferrets. Below each pie graph, RNA profiles of the relative composition of transcripts are shown. Source data are provided as a Source Data file.

present 8 days after pdmCA09, but fully resolved by 12 dpi (Fig. 4f). Infection of ferrets without prior influenza history with recCDV NΔ425-479 did not cause pulmonary edema. In contrast, pdmCA09-experienced animals developed pneumonia that was first detectable 12 days after CDV and rapidly advanced to major hemorrhage 14–19 dpi (Fig. 4f; Supplementary Movie S3, S4; Supplementary Fig. S15).

Histopathology of lung tissue extracted in repeat studies when vehicle-treated pdmCA09 and recCDV NΔ425-479-infected animals became moribund (15–19 dpi) revealed widespread edema, necrosis, and vasculitis in the small airways (Fig. 4g; Supplementary Fig. S16). Lung samples extracted 19 dpi from ferrets that were infected only with recCDV NΔ425-479 contained areas of inflammatory cellular infiltrates, bronchial epithelial hyperplasia and epithelial sloughing consistent with mild interstitial viral pneumonia, but showed no signs of edema and vasculitis, reflected by significantly lower pathology scores than those of the consecutively-infected group (Supplementary Fig. S17). Lung tissues of CDV-vaccinated consecutively infected animals were unremarkable, their pathology scores resembling those of mock-infected ferrets. Gram-staining of tissue samples from consecutively-infected animals showed fulminant bacterial superinfections that were absent from animals infected with CDV only, which clinical microbiology identified as part of the commensal microbiome (Fig. 4h; Supplementary Fig. S17a, b). Metagenomics furthermore revealed a relative expansion of the Escherichia population in the lung microbiome 10 dpi with CDV in both singly and consecutively infected animals (Fig. 4i; Supplementary Dataset S2).

To assess whether priming for exacerbated disease is correlated with severity of the initial viral pneumonia, is strictly dependent on the 28-day interval between infections, or is IAV-specific, we treated the initial pdmCA09-infected animals with the broad-spectrum nucleoside-analog antiviral 4'-fluorouridine (4'-FlU; EIDD-2749)[23], infected animals with pdmCA09 67 days before CDV, or inoculated ferrets with RSV-A2-L19F[24] instead of pdmCA09, respectively (Fig. 5a). Bronchoalveolar lavages (BALs) were sampled at 5 predefined time points throughout the study. Consistent with previously demonstrated α-IAV efficacy of 4'-FlU[25], once daily (q.d.) oral administration at 2 mg/kg bodyweight initiated 24 h after infection with pdmCA09 statistically significantly shortened viral shedding and alleviated clinical signs (Supplementary Fig. S18a, b). However, pharmacological mitigation of IAV disease did not significantly alter viremia titers or virus shedding after subsequent infection with recCDV NΔ425-479 (Fig. 5b, c) and the majority of animals succumbed to hemorrhagic pneumonia (Fig. 5d). An expanded 67-day interval between IAV and CDV infections had no major alleviating effect on CDV disease presentation and hemorrhagic pneumonia outcome (Fig. 5e–g).

Inoculation of ferrets with RSV-A2-L19F resulted in productive infection, characterized by efficient RSV shedding into nasal lavages (Supplementary Fig. S19a), but animals did not display overt clinical signs (Supplementary Fig. S19b). No cross-protective immunity with paramyxovirus infection was observed when we inoculated ferrets 28-days after RSV with recCDV NΔ425-479 (Supplementary Fig. S19c–e), and the majority of animals again succumbed to hemorrhagic pneumonia (Supplementary Fig. S19f).

## Differential expression of TFF peptides after CDV infection as a consequence of prior disease history

RNA-qPCR analysis of BAL samples taken during and after IAV infection of ferrets for expression levels of selected pro-inflammatory and wound-healing cytokines relative to those prior to exposure suggested that respiratory tissues of IAV-experienced animals may be in an anti-inflammatory state at the time of CDV infection compared to immune homeostasis of IAV-naïve animals (Supplementary Fig. S20).

Comparative RNAseq screening of BAL samples extracted 5 dpi from ferrets infected with recCDV NΔ425-479, or consecutively infected with pdmCA09 followed by recCDV NΔ425-479, identified 9 genes as significantly differentially expressed between study groups (Fig. 5h). These included all three genes encoding trefoil factor (TFF) peptides (Fig. 5i), which are known to be involved in protection and repair of gastrointestinal[26] and, although less well studied, respiratory[27] epithelia. Follow-up RT-qPCR of lung tissue extracts harvested 10 days after CDV infection confirmed that TFF-encoding mRNAs were upregulated several 100-fold (TFF1 and 2) to 1000-fold (TFF3) in animals infected only with CDV compared to the consecutively infected groups (Fig. 5j). pdmCA09 infection alone resulted in only slightly increased TFF1 and 2 expression 10 dpi and TFF3 was not upregulated (Supplementary Fig. S21a–c), indicating that extreme induction of TFF1 and 3 is morbillivirus-specific rather than a general response to respiratory virus infections. In the mouse respiratory tract, TFF1 and 3 reportedly colocalize with mucins Muc5AC and Muc5B[28]. However, RT-qPCR did not reveal an equivalent differential upregulation of either Muc5 expression level in the different infection groups (Supplementary Fig. S21d, e).

These results demonstrated that immune priming through unrelated primary viral pneumonia sets the stage for exacerbated subsequent morbillivirus disease. CDV infection during, or shortly after, the recovery phase of an inflammatory episode alters the quality of the host defense to morbillivirus disease, resulting in significantly lower expression of respiratory epithelium protective TFFs, which emerged as correlative for risk to advance to hemorrhagic pneumonia.

## Late onset treatment of CDV infection does not mitigate signs of CDV disease, but alters lethal outcome

To assess the effect of treatment of morbillivirus infection on disease outcome, we again consecutively infected animals with pdmCA09 (Supplementary Fig. S22) followed by recCDV NΔ425-479, and initiated oral treatment with GHP-88309 5, 7, 10, and 14 dpi with CDV (Fig. 6a). Consistent with results of the previous GHP-88309 treatment studies, PBMC-associated CDV viremia was mitigated only when treatment was initiated 5 dpi (Fig. 6b). All animals started on GHP-88309 5 or 7 dpi with CDV survived, whereas vehicle-treated animals succumbed to hemorrhagic pneumonia between days 14–19 after CDV infection (Fig. 6c). A majority of animals in the 10 dpi treatment group recovered, but most ferrets first treated with GHP-88309 14 dpi with CDV developed lethal pneumonia. Lymphocyte counts were fully preserved in CDV-vaccinated control animals and ferrets of the 5 dpi treatment group, and were partially preserved in animals of the 7 dpi group (Fig. 6d). Although later treatment start did not mitigate lymphocytopenia, PBMC repopulation was accelerated in the 10 dpi treatment group. Circulating granulocyte populations did not deviate from the

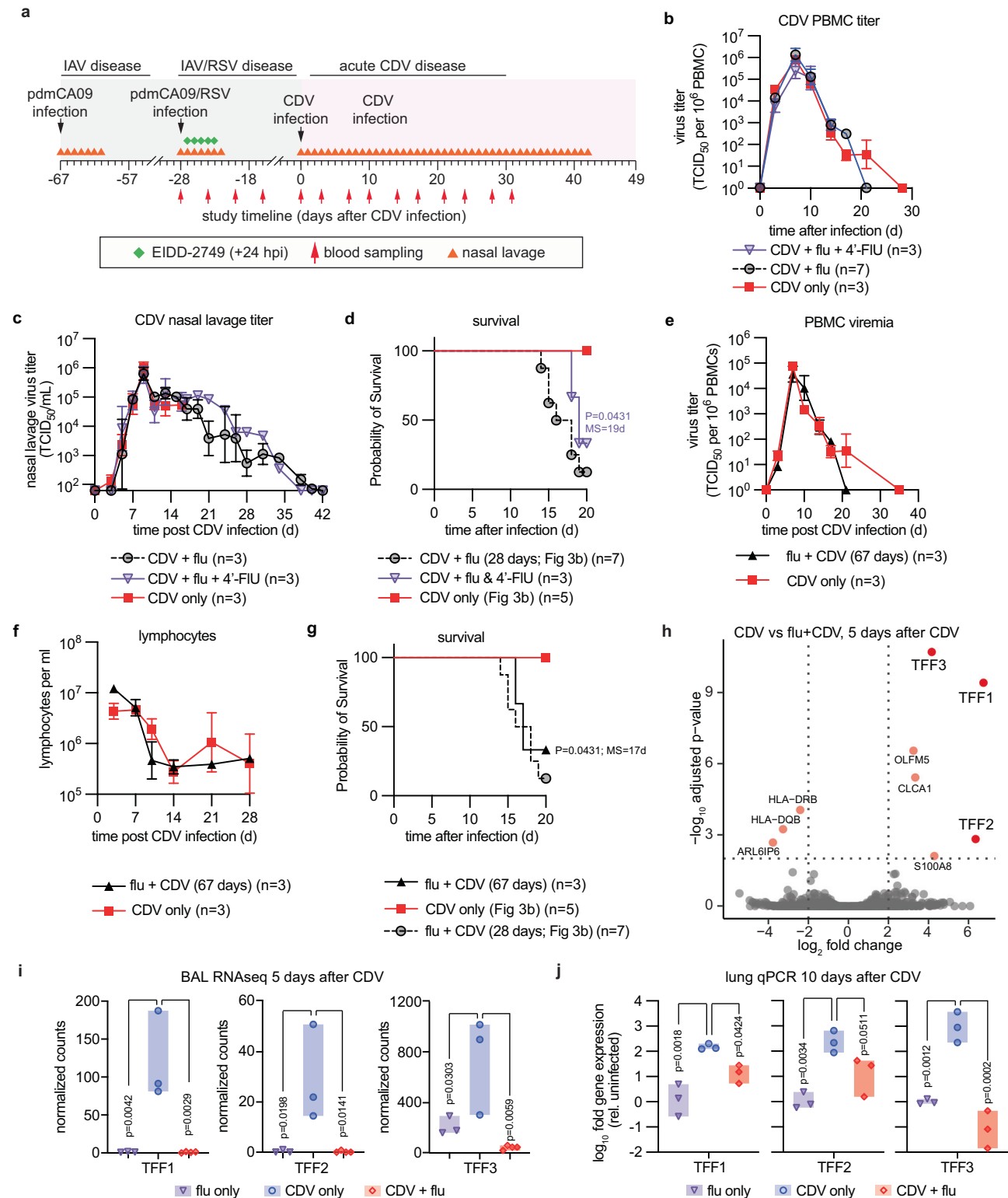

**Fig. 5 | Disease history affects CDV-induced TFF upregulation. a** Schematic of study design. **b, c** PBMC-associated CDV viremia (**b**) and nasal lavage (**d**) titers of ferrets infected with recCDV-5804p NΔ425-479. **d** Survival of consecutively IAV + CDV-infected ferrets after treatment of primary IAV with 4-FlU. **e–g** PBMC-associated primary viremia titers (**e**), circulating lymphocyte counts (**f**), and survival (**g**) of ferrets infected with recCDV-5804p NΔ425-479 67 dpi with primary pdmCA09. Symbols represent geometric means ± geometric SD (**b, c, e, f**), lines intersect means; log-rank (Mantel-Cox) test, median survival is stated (**d, g**); n numbers as specified. **h** RNAseq screen of differentially expressed transcripts present in BAL fluids extracted 5 dpi with CDV from consecutively infected versus CDV-only ferrets. Differentially expressed genes were determined with the Wald

test and P values corrected for multiple tests using the method of Benjamini and Hochberg. Significance thresholds (adjusted $p < 0.01$, log2 fold change > 2) are shown as dotted horizontal and vertical lines, respectively. **i** Trefoil factor (TFF1, TFF2, and TFF3) transcripts in BAL fluids harvested 5 dpi with CDV from consecutively infected versus CDV-only ferrets. **j** RT-qPCR quantitation of relative presence of trefoil factor-encoding message in lung tissue of consecutively infected versus CDV-only ferrets, extracted 10 dpi with CDV. Symbols (**i, j**) represent individual animals, bars show range; 1-way ANOVA with Dunnett's post-hoc test (**i, j**); $n = 3$. Source data are provided as a Source Data file.

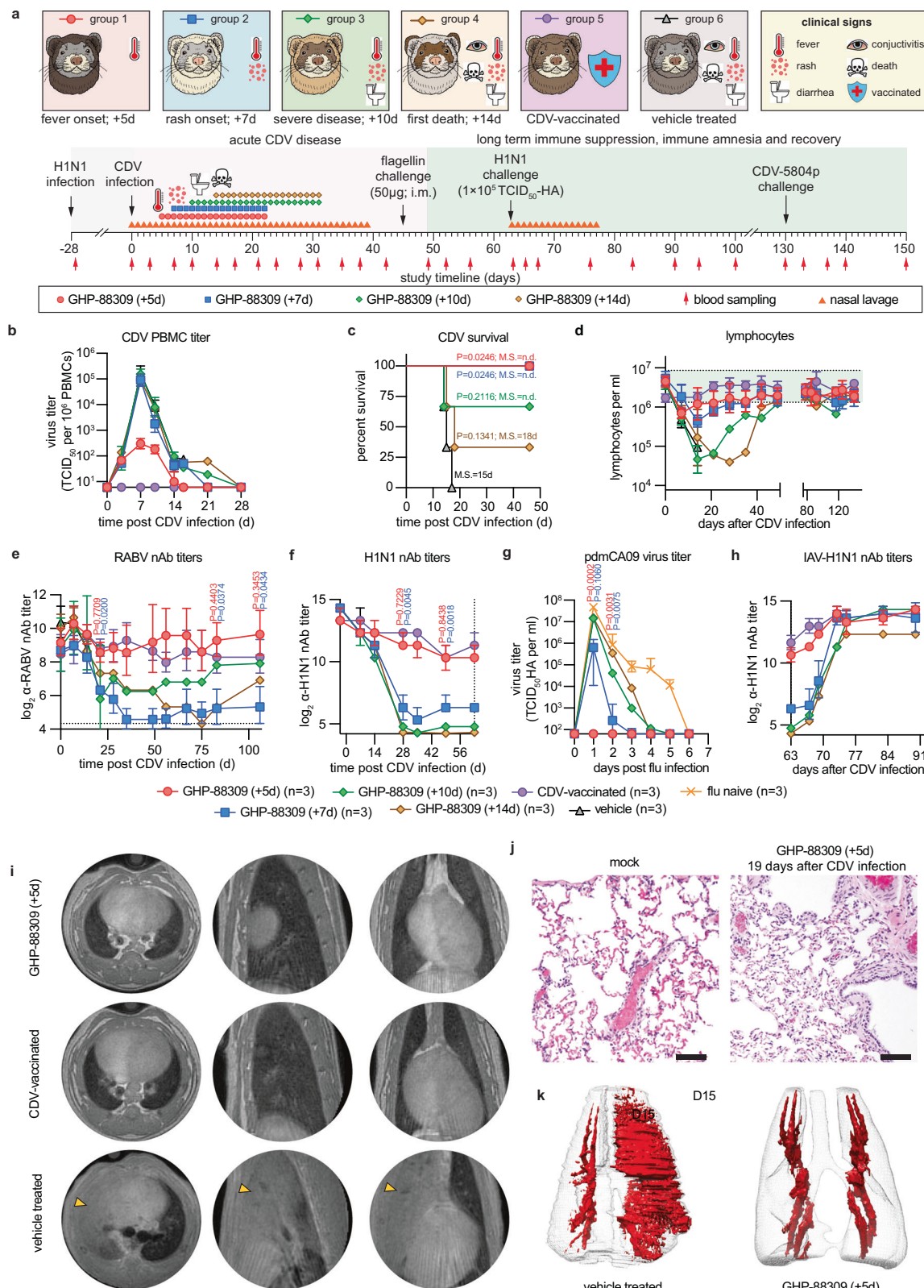

normal range in any study group, making granulocytes transiently the predominant white blood cell population in animals experiencing lymphocytopenia (Supplementary Fig. S23). nAb titers against RABV and IAV H1N1 reflected the lymphocytopenia results; fully preserved in vaccinated and 5 dpi GHP-88309-treated animals, partially preserved in the 7 dpi treatment group, and not preserved when GHP-88309 was initiated 10 or 14 dpi with CDV (Fig. 6e, f). Accordingly, only ferrets in

the 5 and 7 dpi GHP-88309 treatment groups were fully or partially, respectively, protected against pdmCA09 challenge 62 days after CDV (Fig. 6g). However, surviving animals of all GHP-88309 treatment groups mounted a robust de novo α-IAV H1N1 nAb response following challenge with pdmCA09 (Fig. 6h). At study end, all ferrets had furthermore developed robust immunity against CDV reinfection (Supplementary Fig. S24).

**Fig. 6 | Effect of therapeutic intervention on severe disease outcomes. a** Study schematic. **b–f** PBMC-associated viremia titers (**b**), survival (**c**), lymphocyte counts (**d**), and RABV (**e**) and IAV H1N1 (**f**) nAb titers of consecutively infected and GHP-88309-treated ferrets from (**a**). Green shading in (**d**) denotes normal range. **g** pdmCA09 nasal lavage titers after IAV challenge of ferrets recovered from CDV disease. **h** IAV H1N1 nAb titers after challenge of ferrets recovered from CDV disease with pdmCA09 on study day 65. **i** MRI slices of consecutively infected and GHP-88309-treated (top row), CDV-vaccinated (middle row), or vehicle-treated (bottom row) ferrets, taken 15 dpi with CDV. Axial, coronal, and sagittal slices of the same lung are shown. Arrow heads denote hemorrhagic infiltrates. **j** 3D MRI reconstructions and segmentations of lungs from consecutively IAV CDV-infected and GHP-88309 or vehicle-treated ferrets. Images were collected 15 dpi with CDV; fluids shown in red. **k** Lung histopathology of consecutively infected and GHP-88309-treated or uninfected (mock) ferrets, assessed 19 dpi with CDV; scale bars represent 100 μm. Symbols (**b**, **d–h**) represent geometric means ± geometric SD, lines intersect means; error was not calculated for later timepoints of the GHP-88309 (+10d) and GHP-88309 (+14d) conditions, when individual animals had reached endpoints and subgroup sizes were below 3. Log-rank (Mantel-Cox) test (**c**), median survival is stated; 2-way ANOVA with Dunnett's post-hoc test (**e–g**); n numbers as specified. Source data are provided as a Source Data file.

Longitudinal MRI analysis 15 dpi with CDV of consecutively IAV and CDV-infected animals treated with GHP-88309 demonstrated suppression of pulmonary edema and hemorrhage (Fig. 6i, j; Supplementary Movies S5, S6), which was confirmed by histopathology of treated animals (Fig. 6k). Again, whole genome sequencing of CDV recovered from GHP-88309-treated ferrets 9–19 dpi revealed no known[12] resistance mutations (Supplementary Dataset S3).

Outcome reversal of lethal hemorrhagic pneumonia through late-onset GHP-88309 established precedent for therapeutic benefit of direct-acting antiviral therapy initiated after the time window for mitigation of primary clinical signs of morbillivirus disease has closed. The results demonstrate that GHP-88309 treatment paradigms for mitigation of morbillivirus-induced immune amnesia are independent of whether immunity was naturally acquired or vaccine-induced.

## Discussion

Reflecting a paucity of effective antiviral therapeutics, their benefit is poorly defined for most pathogenic paramyxoviruses and unknown for members of the Morbillivirus genus, specifically. Recent work testing remdesivir therapy in a non-human primate model of MeV infection reported transient reduction of viral RNA through post-exposure prophylactic treatment, but lymphocytopenia and clinical disease parameters were not improved and virus replication rebounded[29]. Since remdesivir was designed to yield high liver exposure[30], sustained inhibitory concentrations may not be reached in MeV-relevant target cells and tissues. Having identified orally efficacious morbillivirus[11,31] and broadened-spectrum paramyxovirus[12] inhibitors, we established in this study treatment paradigms for morbillivirus disease. Results support five major conclusions: i) efficacious antivirals such as GHP-88309 expand the therapeutic time window to eight days compared to therapeutic vaccination; ii) infections with engineered attenuated recCDV confirm the morbillivirus immune amnesia hypothesis in the ferret model; iii) therapeutic mitigation of primary clinical signs of morbillivirus disease and immune amnesia is beneficial when treatment is initiated before, or at peak, of primary viremia; iv) prior disease history of the host defines the risk for exacerbated, fatal morbillivirus infection; and v) late-onset anti-morbillivirus treatment prevents lethal bacterial superinfection.

The impact of prior infection on the outcome of morbillivirus infection was most unexpected. Although heightened long-term susceptibility to secondary infections after severe primary viral or bacterial pneumonia was described in mice[32], this sepsis-induced transient immunosuppression was attributed to poor antigen-presentation capacity of DCs and AMs in the TGF-β-driven anti-inflammatory microenvironment established during the tissue healing phase after resolution of the primary infection[33]. However, immune priming for catastrophic morbillivirus disease did not require severe primary pneumonia, since highly efficacious treatment of pdmCA09 infection with 4′-FlU[25] did not prevent hemorrhagic pneumonia after subsequent CDV infection. Unlike therapeutic intervention with the primary IAV infection, late-onset treatment of the morbillivirus infection with GHP-88309 initiated shortly before uncontrolled amplification of commensal bacteria changed outcome, indicating that CDV replication in the respiratory epithelium after basolateral re-invasion of lungs is instrumental for progression to lethal secondary complications. Our study revealed that morbillivirus triggers massive upregulation of TFF1, TFF3 and, to a lesser degree, TFF2 expression, provided infection occurred during lung immune homeostasis. In the respiratory tract, TFFs are predominantly produced by glandular and mucus-secreting epithelial and hematopoietic cells[27,34–36], and TFF1 and 3 specifically enhance mucociliary bacterial clearance through increasing mucus viscosity[27,37,38].

Based on these observations, we propose an immune-priming-based mechanism leading to exacerbated disease. IAV infection reportedly results in transient population of the respiratory tract with functionally altered DCs and AMs to restore immune homeostasis after the pro-inflammatory response, which are unable to clear bacterial pathogens efficiently[33]. CDV infection during the recovery phase, followed by basolateral invasion of the respiratory epithelium after primary viremia, further depletes DCs and AMs[6] and fails to upregulate TFF expression, setting the stage for catastrophic bacterial pneumonia through impaired clearance of commensal bacteria. How does prior IAV infection affect TFF1 and 3 upregulation? Type 2 cytokines IL-4 and IL-13 stimulate TFF3 synthesis in the GI tract in a STAT6-dependent process[27]. Upregulation of immunosuppressive cytokines such as TGFβ and downregulation of pro-inflammatory IL-1β at the time of CDV infection after influenza may create an environment incompatible with TFF induction after CDV, but regulation of TFF3 expression in the airway epithelium is overall poorly understood. In mice, IAV stimulates lung-resident lineage-negative epithelial progenitor (LNEP) cells that are the major responders in distal lung after tissue damage[39] and express high levels of TFF2[27] potentially creating a negative feedback loop that prevents TFF1 and 3 upregulation. Future work must focus on the regulation of TFF3 levels in the homeostatic respiratory tract to better understand the molecular basis for long-lasting impaired expression after influenza virus priming.

It is currently unknown whether MeV equally induces TFF1 and 3 expression in the human respiratory tract and what impact disease history of measles patients may have on severity of secondary complications. Anecdotal cases of hemorrhagic pneumonia associated with measles have been reported[40,41], but prior disease history of the patients was not documented. However, bacterial superinfection after measles such as laryngitis, bronchitis, and otitis media are common[9]. Typically attributed to impaired adaptive immunity due to lymphocytopenia, our ferret data support that respiratory disease history of measles patients should be considered as risk factor for advance to severe bacterial superinfections. Limitations affecting the predictive power of the CDV ferret model for human measles include different disease dynamics[42], higher mortality rates of the attenuated recCDV NΔ425-479, higher inherent neurotropism of CDV than MeV[42], and untested cross-species consistency of pharmacokinetic and efficacy performance of GHP-88309.

This study establishes treatment paradigms for efficacious pharmacological intervention in morbillivirus disease, defines respiratory disease history as a correlate for the risk of severe bacterial superinfection, and provides precedent for therapeutic benefit of treatment

of an acute RNA virus infection with direct-acting antivirals initiated after the window for mitigation of primary clinical signs has closed.

## Methods

### Study design
The objectives of this study were to determine the efficacy of GHP-88309 against morbillivirus disease in the CDV-ferret animal model and identify therapeutic windows for the successful treatment of a lethal morbillivirus infection. The CDV-ferret model was chosen because it provides a surrogate animal model of severe morbillivirus disease, including many of the hallmark symptoms and clinical signs of measles virus in humans. Female ferrets (Mustela putorius furo), family mustelids, genus Mustela, 6–10 months of age, were used for all experiments. Group sizes were three individual animals per conditions unless otherwise stated. The effect of treatment on virus replication and clinical signs in ferrets was determined using multiple therapeutic treatment regimens. Treatment was considered efficacious when statistically significant reductions in viremia and shed virus titers in PBMCs and nasal lavages, respectively, were observed and the duration and severity of clinical signs and immune suppression were decreased. Study endpoints were predefined prior to initiating experiments. At least two ferrets were used in all PK and tolerability studies. Groups of at least three ferrets were used in all in vivo efficacy studies. Before initiating experiments, animals were randomly assigned into groups. Exact numbers of independent biological repeats (individual animals) for each experiment are specified in the figures or figure legends. All quantitative source data are provided in the Source Data file.

### Cell lines and transfections
MDCK cells (American Type Culture Collection, CCL-34), Human carcinoma (HEp-2, American Type Culture Collection, CCL-23), and African green monkey kidney epithelial cells (American Type Culture Collection CCK-81) stably expressing canine signaling lymphocytic activation molecule (Vero-cSLAM[43]) were maintained at 37 °C and 5% $CO_2$ in Dulbecco's modified Eagle's medium (DMEM) supplemented with 7.5% fetal bovine serum (FBS). All immortalized cell lines used in this study were regularly tested for microbial contamination (6-month intervals).

### Viruses
recCDV-5804p, recCDV-5804p-Nectin-4-blind-eGFP, and recCDV-5804p-NΔ425-479 stocks were propagated on Vero-cSLAM and titrated by $TCID_{50}$ assay. recRSV-A2-L19 stocks were propagated on HEp-2 cells and titrated by $TCID_{50}$ assay. A/California/7/2009 (H1N1) and A/Wisconsin/67/2005 (H3N2) were propagated on MDCK cells for 2 to 3 days at 37 °C. Influenza viruses were titrated by $TCID_{50}$-hemagglutination ($TCID_{50}$-HA) assay on MDCK cells[44].

### Viral Whole Genome Sequencing
For whole genome sequencing of recCDV-5804p-NΔ425-479 stocks and ferret nasal lavages or BALF following established methods[12], genomic DNA was first depleted from the RNA extracts with Turbo DNAse (Thermo #AM1907). First-strand cDNA synthesis was performed using SuperScript IV (Thermo) and random hexamers (Thermo) followed by second strand synthesis with Sequenase 2.0 kit (Thermo). Double-stranded cDNA was simultaneously fragmented and barcoded using Nextera chemistry (Nextera XT and Nextera Flex). The resulting libraries were pooled at equal concentrations, and average library size was determined using Agilent DNA D1000 Tape Station kit (Agilent). Pooled libraries were sequenced on a Nextseq 500 or Novaseq 6000. Reads were quality and adapter-trimmed with fastp[45]. The 3 leading bases, 3 trailing bases, and bases with mean phred< 20 in a 4-bp sliding window were cut from each read. Reads shorter than 45 bp and reads in which > 40% bases had phred < 15 were removed.

Consensus genomes for the recCDV-5804p-NΔ425-479 stocks were generated from the trimmed reads using REVICA (https://github.com/greninger-lab/revica) with the Canine distemper virus strain 5804 P (GenBank #AY386316.1) as the initial reference. The LAVA pipeline, available at https://github.com/greninger-lab/lava/tree/Rava_Slippage-Patch, was used to interrogate allele frequency changes from the 425 input consensus genome[12]. Samples with < 10,000 mapped reads or < 95% coverage breadth were excluded from variant analysis. Interactive HTML plots highlighting viral allele frequencies relative to inoculum are included as (Supplementary Datasets 1 and 3). Sequence reads have been uploaded to the sequence read archive under a.

### Virus yield reduction
For virus yield-based dose–response assays, cells were infected (M.O.I. = 0.01 $TCID_{50}$ units per cell) in a 24-well plate format with recCDV-5804, recCDV-5804p-NΔ425-479, or drug resistant recombinants, in the presence of serial compound dilutions. Cell-associated progeny virus was harvested 48 h after infection. Viral titers were determined through $TCID_{50}$ titration. Four-parameter variable slope regression modeling was used to determine $EC_{50}$ and $EC_{90}$ concentrations.

### Pharmacokinetics studies in ferrets
Female ferrets (6 to 10 months of age) received from Triple F Farms were rested for 1 week, randomly assigned to study groups, and dosed orally with GHP-88309 dissolved in 1% methylcellulose. Blood was collected from the anterior vena cava and tissue sampling at the specified time points. Two to three animals per group were sampled for PK analyses. Plasma was separated from blood in microvette CB300 EDTA tubes (2000 × g; 5 min; 4 °C) (Sarstedt Inc) and tissue samples were snap frozen and stored at −80 °C prior to analysis by LC-MS/MS. The calibration curve range was 10–100,000 ng ml$^{-1}$ in blank plasma for single-dose PK and 10 ng/ml to 50,000 ng/ml for all other studies. Quality control samples of 30, 300, 7500, and 50,000 ng/mL of blank plasma were run before and after the single-dose PK samples and quality control samples of 30, 750, and 5000 ng/ml were run before and after all other plasma samples. The calibration curve range was 1.00–2000 ng/ml of tissue lysate for blank tissues. Quality control samples of 3, 30, and 600 ng/ml of tissue lysate in blank tissues were analyzed at the beginning of each sample set. Calibration in each matrix showed linearity with $R^2$ values of > 0.99.

### RSV and IAV infections of ferrets
Female ferrets were purchased from Triple F Farms and housed in an ABSL-2 facility. Prior to study start, ferrets were rested for 1 week, then randomly assigned to study groups. Animals infected with RSV-A2-L19 were in a separate ABSL2 room from influenza infected ferrets. For influenza virus and RSV infections, ferrets were anesthetized using dexmedetomidine/ketamine. Anesthetized ferrets were inoculated intranasally with $1 \times 10^5$ $TCID_{50}$ units of IAV-A/CA/07/2009 (H1N1) or $1 \times 10^6$ TCID50 units of RSV-A2-L19 in a volume of 300 μl per nare. Clinical signs (bodyweight and temperature) were monitored once daily. Nasal lavages were performed once daily using 1 ml of PBS containing 2× antibiotics-antimycotics (Gibco). Treatment with 4'-FlU through oral gavage was administered using a once daily (q.d.) regimen.

### CDV infections of ferrets
Female ferrets were received from Triple F Farms and housed in an ABSL-2 facility. Ferrets were rested for one week prior to the start of each study and randomly assigned into groups. Influenza vaccinations (quadrivalent, Flucelvax; Seqirus, Inc.) were administered at specified time points prior to CDV infection. Prior to infection, ferrets were anesthetized with dexmedetomidine/ketamine, and infected intranasally with $2 \times 10^5$ $TCID_{50}$ of recCDV-5804p, recCDV-5804p-Nectin-4-

blind-eGFP or recCDV-5804p-N$\Delta$425-479 in 200 μl (100 μl per nare). Twice daily treatment with GHP-88309 was initiated at specified times after infection (3, 5, 7, 10, or 24 days after infection). GHP-88309 was administered by oral gavage in 3.5 ml 1% methylcellulose and flushed with 3.5 ml high-calorie liquid dietary supplement. Control groups were administered equivalent volumes of 1% methylcellulose. Body-weight and temperature were monitored daily. Additional monitoring for clinical signs and any potential adverse effects was performed daily. Blood was harvested at specified time points. To assess viremia, PBMCs were isolated from blood by Ficoll gradient centrifugation. Viremia titers were determined by coculturing serial dilutions of purified PBMCs with Vero-cSLAM cells and expressed as $TCID_{50}$ per $10^6$ PBMCs. Complete blood count (CBC) analyses were performed using a VetScan HM5 (Abaxis) in accordance with the manufacturer's protocol. Shed viral loads were determined from nasal lavages and titrated by $TCID_{50}$. For studies involving influenza challenges, surviving CDV-infected ferrets were anesthetized with dexmedetomidine/ketamine, and infected intranasally with $1 \times 10^5$ [recIAV-A/California/07/2009 (H1N1)] or $2 \times 10^5$ $TCID_{50}$HA [A/Wisconsin/67/2005 (H3N2)] in 200 μl (100 μl per nare). Animals were monitored once daily for clinical signs, including loss of bodyweight and fever. Upper respiratory virus load was measured from nasal lavages performed once daily for up to 7 days post influenza infection. Rash was scored using a scale of 0-2 (0, no rash; 1, mild and localized rash; 2, severe and widespread rash). Diarrhea was scored using a scale of 0–2 (0, normal stool; 1, soft stool; 2, runny stool and/or rectal inflammation). Conjunctivitis was scored using a scale of 0-2 (0, clear eyes; 1, puffiness or mild swelling in at least one eye; 2, presence of puss in both eyes or unable to open eyes without cleaning).

### Systemic interferon and cytokine profiling

The relative expression of interferon, cytokines, and interferon-stimulated genes (ISGs) was determined by real-time PCR analysis. RNA was isolated from purified PBMCs that were harvested at different time points after infection. Complementary DNA was synthesized by reverse transcription using SuperScript III (Invitrogen) using oligo-dT primers in accordance with the manufacturer's protocol. Real-time PCR was performed using Fast SYBR Green Master Mix (Applied Biosystems) on a Quantstudio 3 real-time PCR system (Applied Biosystems). Values were normalized to glyceraldehyde-3-phosphate dehydrogenase mRNA, analyzed by the comparative threshold cycle ($\Delta\Delta C_T$) method and expressed relative to mock-infected animals. Sequences of primers are shown in Supplementary Table S3.

### Determination of neutralizing antibody titers

Plasma was heat inactivated and serially diluted (twofold steps) in serum-free DMEM, mixed with 100 $TCID_{50}$ units of recCDV-5804p, recIAV-A/California/07/2009 (H1N1), or VSV-RABV-G and incubated for 1 h at 37 °C. Mixtures were transferred to cell monolayers and virus neutralization was measured after 3 d by visualization of syncytia, hemagglutination assay, or GFP-positive infected cells for recCDV-5804p, recIAV-A/California/07/2009 (H1N1), and VSV-RABV-G, respectively. Equal amounts of virus in FBS, DMEM containing 7.5% FBS, and serum-free DMEM served as controls. Each blood sample was tested in two technical repeats.

### Quantitation of CDV N protein-encoding RNA in PBMCs

CDV N protein-encoding RNA was detected using the a primer-probe set cdv_n_taq_fw, cdv_n_taq_rev, and cdv_n_probe. RT-qPCR reactions were performed using a QuantStudio 3 real-time PCR system and the QuantStudio Design and Analysis package (version 1.5.2). The CDV N primer-probe set was used with Taqman Fast Virus 1-step master mix (Thermo Fisher Scientific) to detect viral RNA. To calculate RNA copy numbers, a standard curve was created using a linearized pTM1-CDV N

plasmid of known concentration as template. Samples were normalized to the numbers of input PBMCs.

### Ferret MRI

All ferrets were imaged on a high-resolution 7 T Bruker (70/20) Biospec MRI scanner at Georgia State University using the ParaVision software package (Bruker, Billerica, MA, version 360.3.4). Animals were anesthetized using a combination of dexamethasone and isoflurane. Respiration rates and body temperature were continuously monitored and maintained using a small-animal physiological monitoring system (SA Instruments Inc, Stony Brook, NY and Kent Scientific, Somnosuite Systems, Torrington, CT). Anesthesia was adjusted to maintain a respiration rate of 40-60 breaths per minute. A 112/86 mm circularly polarized transmitter/receiver coil was employed for in-vivo imaging. Following the localizer scan to position the animal in the center of the magnet, two types of MR protocols were used to capture the lungs in coronal and axial orientations. To overcome the limitations imposed by short T2* of lungs and to reduce motion, a 3D Ultra short echo time (3D UTE) was used to acquire high-quality images with the following parameters: MR acquisition parameters include: 4.3 μs RF block pulse; flip angle ($\alpha$) = 3.9°; 51,360 radial projections; 128 points on free induction decay (FID); field-of-view (FOV) = 58 × 54 × 67 (mm³); image matrix size = 128 × 128 × 128 (voxels3); receiver bandwidth (BW) = 200 kHz; TE = 0.06 (ms); repetition time (TR) = 4.206 (ms) and 2 signal averages. The total acquisition time for an individual ferret UTE scan was ~7 minutes. An additional 2D T1 IG FLASH (intragate fast low angle shot) self-gated MRI was used to collect images along coronal direction with the following parameters: TR/TE = 400/3 msec, flip angle = 30°, oversampling = 10, 30 slices, matrix size 140 × 128, FOV = 70 mm × 64 mm. The total acquisition time was 8 min. The images were converted into DICOM/Nifti formats using Bruker PV-360 software and reconstructed and processed using ImageJ (version 2.9.0) and ITK-SNAP (version 3.6.2-alpha).

### Histopathology

Pathology scoring was performed according to the following scale: for alveoli, bronchiolitis, and pleuritis scores, scoring was based upon distribution: 0 = no lesions, 1 = focal, 2 = multifocal, 3 = multifocal to coalescing, 4 = diffuse; for perivascular cuffing (PVC) score: 1 = 1 layer of leukocytes surrounding most affected vessel, 2 = 2–5 layers, 3 = 6–10 layers, 4 = more than 10 layers; for vasculitis score: 1 = infiltration of vessel wall by leukocytes, 2 = infiltration and separation of smooth muscle cells by edema, 3 = same changes as 2 with fibrinoid change, 4 = effacement of the vessel wall; for interstitial pneumonia score: 1 = infiltration of alveolar septa by 1 leukocyte layer thickness, 2 = expansion by 2 leukocyte thickness, 3 = 3 leukocytes thick, 4 = 4 leukocytes thick or more. The sum of individual scores represents a total histopathology score, generated for each animal.

### RNAseq

Total RNA was treated with Turbo DNAse (ThermoFisher) and used as input for the Illumina Stranded Total RNA (Ribo-Zero Plus Microbiome) library preparation kit using the manufacturer's specifications, then sequenced on an Illumina NovaSeq 6000 to obtain approximately 25 million paired end 150 bp reads. Raw reads were quality- and adapter-trimmed with Trimmomatic v0.39[46]. Metagenomic analysis of trimmed read pairs was performed using the CZID pipeline[47–49] and host and CDV-specific proportions determined. Trimmed read pairs were pseudoaligned to the draft ferret transcriptome MusPutFur1.0, INSDC Assembly GCA_000215625.1[50] using Kallisto v0.46[51] and transcripts aggregated by gene. Genes with an average raw expression level less than 1 raw count per sample were filtered prior to analysis. Differential expression between groups of CDV-infected and flu-pretreated CDV-infected ferret samples collected during survival BAL procedures was calculated with the Wald test in DEseq2[52], with an

adjusted p value significance threshold of 0.01. All analyses were performed using R v4.2.1. Raw sequence reads used for RNAseq analysis have been uploaded to the sequence read archive, BioProject PRJNA1004336.

## Inclusion and ethics statement

All animal work was performed in compliance with the Guide for the Care and Use of Laboratory Animals of the National Institutes of Health and the Animal Welfare Act Code of Federal Regulations. Experiments involving ferrets were approved by the Georgia State University IACUC under protocols A22035 and A18035. All ferret studies were carried out with female animals only and no sex-based analyses have been performed for the following reason: per IACUC protocol, co-housing studies may not be carried out with male ferrets, since males are territorial and combative when co-housed, resulting in severe injury or death from fight wounds that would require termination of the study. Vivarium capacity for long-term housing of large animals in ABSL-2 containment does not permit the use of singly-housed male ferrets. All experiments using infectious CDV, RSV IAV, and VSV-deltaG were approved by the Georgia State Institutional Biosafety Committee under protocol B21029 and performed in BSL-2/ABSL-2 facilities at Georgia State University.

## Statistics and reproducibility

One-way or two-way analysis of variance (ANOVA) with Dunnett's or Sidak's multiple comparison post-hoc tests, was used to assess statistical differences. All statistical analyses were carried out in GraphPad Prism software (Version 9.3.1). Specific statistical tests applied to individual data sets are specified in the corresponding figure legends. The number of individual biological replicates for all graphical representations are shown in the figures. Representations of mean ± SD or median ± 95% CI of experimental uncertainty are shown and specified in the figure legends. Fourteen-day tolerability studies were based on two ferrets. All statistical analyses and exact $P$-values are shown in the Supplementary Dataset S4. Alpha levels were set to 0.05 for all significance analyses.

## Reporting summary

Further information on research design is available in the Nature Portfolio Reporting Summary linked to this article.

## Data availability

The amplicon tiling sequencing reads generated in this study have been deposited in the NCBI BioProject database under accession code PRJNA1004336. All other data generated in this study are provided in the Supplementary Information, Supplementary Datasets S1–S4, and the Source Data file. Quantitative source data have been deposited in Figshare (https://doi.org/10.6084/m9.figshare.24076626). Source data are provided with this paper.

## Code availability

Sequencing reads were analyzed using the TAYLOR pipeline, available at https://github.com/greninger-lab/covid_swift_pipeline (https://doi.org/10.5281/zenodo.6142073). All commercial computer codes and algorithms used are specified in Methods.

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

## Acknowledgements

We thank the Georgia State University Department of Animal Resources, the Georgia State University Advanced Translational Imaging Facility (ATIF), and the University of Georgia Pathogenesis Core for expert assistance. The VSV-ΔG and recCDV-5804p genomic plasmids and Vero-canine SLAM cells were kind gifts of M.J. Schnell, V. von Messling, and Y. Yanagi, respectively. This study was supported, in part, by public health service grants AI071002 (to R.K.P.) and AI171403 project 3 (to R.K.P.) and AI171403 Scientific Core D (to A.L.G.) from the NIH/NIAID.

## Author contributions

R.K.P. and R.M.C. conceived and designed the experiments. R.M.C., J.D.W., C.M.L., H.-J.K., J.-J.Y., and R.K.P. conducted most of the experiments. R.M.C. created all figure schematics. A.L.G., N.A.L., and E.B.S. performed next-generation sequencing. Z.M.S., M.K.A., R.E.K., and A.A.K. performed mass spectrometry analysis. K.H. performed MRI imaging. K.S. performed a histopathology analysis. M.G., M.G.N., A.T.G., and R.L.d.S. provided critical materials. R.M.C., J.D.W., C.M.L., N.A.L., A.L.G., and R.K.P. analyzed the data. R.K.P. and R.M.C. wrote the manuscript.

## Competing interests

R.K.P. and R.M.C. are co-inventors on a patent filing covering method of use of GHP-88309 for antiviral therapy. This study could affect their personal financial status. R.K.P. reports contract testing from Enanta Pharmaceuticals and Atea Pharmaceuticals, and research support from Gilead Sciences, outside of the described work. R.M.C. reports consulting for Merck & Co., outside of the described work. A.L.G. reports contract testing from Abbott, Cepheid, Novavax, Pfizer, Janssen and Hologic and research support from Gilead Sciences, outside of the described work. R.L.d.S. reports research support from Gilead Sciences and Themis Biosciences, outside of the described work. All other authors declare that they have no competing interests to report.

## Additional information

[1]Center for Translational Antiviral Research, Georgia State University Institute for Biomedical Sciences, Atlanta, GA 30303, USA. [2]Virology Division, Department of Laboratory Medicine, University of Washington, Seattle, WA 98195, USA. [3]Emory Institute for Drug Development, Emory University, Atlanta, GA 30322, USA. [4]Department of Viroscience, Erasmus MC, Rotterdam, Netherlands. [5]Advanced Translational Imaging Facility, Georgia State University, Atlanta, GA 30303, USA. [6]Department of Pathology, College of Veterinary Medicine, University of Georgia, Athens, GA 30602, USA.
✉e-mail: rplemper@gsu.edu

