## [Peer Review File · Nature Communications]

Therapeutic mitigation of measles-like immune amnesia and exacerbated disease after prior respiratory virus infections in ferretsReviewers' Comments:

Reviewer #1:

Remarks to the Author:

In this paper Cox et al used canine distemper virus (CDV) infection in ferrets to model measles virus (MeV) infection. The study demonstrates the effectiveness of broad-spectrum paramyxovirus inhibitor GHP-88309 in multiple ways. Firstly, it demonstrates the effectiveness of this drug in reducing CDV lethality. Secondly, it demonstrates its effectiveness in preventing CDV induced immune amnesia and lastly, the drug mitigates exacerbated diseases outcomes from secondary CDV infection following primary influenza infection. The study was very thoroughly done and have explored several valuable questions pertaining to morbillivirus disease. However, the presentation of the data can be a little difficult to follow. Overall, highly valuable work that should be ready for publication with a few tweaks.

There are a few minor comments below that can help improve the clarity of this paper:

- a) Clinical scores are recorded in some figures (extended figure 2 etc.) however there is no scoring rubric described in the method. How was severity of rash, conjunctivitis and diarrhea ascertained? Some description of scores would improve the interpretation of this figure.
- b) Supplementary figure 1d: the last column on right of the bar graph has no label
- c) Figure 1a: the symbols next to the animals in each box presumably shows clinical signs? The vehicle treated animal have no symbols next to it though, even though they presented all symptoms? This is not very clear overall.
- d) Line 71, please replace 'Neither' with 'neither'
- e) Figure 2e: what do the symbols on top of the timeline mean? The nasal lavage symbol makes sense, but the other shapes are not clearly explained anywhere.
- f) Figure 3: This figure and section of the study was least clear. How many groups are there? What does CDV naïve/no flu and CDV naïve/flu mean? Why are there other groups such as CDV vaccinated/flu in extended figure 6 but not in figure 3? schematic of the groups like in Figure 1 or 2 would have helped with this. Moreover, in figure 3F, what does the two sections (influenza and CDV disease) mean on top for the CDV only group? Does this mean no prior influenza infection was done? Again, lack of clear description of how many groups there are makes this very confusing to interpret.
- g) Figure 3F: The disappearance of Rodentibacter at 0dpi compared to -28 and -10 dpi stands out distinctly but is not commented on. Any thoughts on this?
- h) Supplementary figure S4a: CDV nectin 4 blind nasal titres are missing. Overall comment about S4 is that data from animals infected with WT CDV is missing and should be included.
- i) Line 129, GHP treatment upto 10dpi reduced shedding in nasal lavages, but it should be noted that titres in PBMCs remain unchanged compared to vehicle only group.
- j) Line 131, Supplementary figure 5a-d deal with CBC in animals infected with different CDV strains and doesn't deal with animals treated with GHP. Needs to be removed from this line and mentioned in relevant earlier sections
- k) Was whole genome sequencing done on only ferret nasal wash samples?
- l) Line 142, The 12 hour data of proinflammatory cytokines is not shown in supplementary figure S6.
- m) Line 153, H1N1 antibody titres are said to be present following infection but no data is shown?
- n) Figure 3h: Was TFF levels measured in RSV+CDV combinations to test the hypothesis that reduced TFF levels are linked to exacerbated disease outcomes?
- o) Supplementary figure S13(a-e): Though some differences can be observed in IAV experienced animals, the data is very variable making it difficult to make reliable conclusions regarding the anti-inflammatory state of animals at time of CDV infection. The differences in fold changes are also quite small (4 fold increase in TGFβ in IAV animal vs 2-fold in CDV only). This data either needs to be tightened with more repeats or the language needs to be softened away from hard conclusions.

Reviewer #2:

Remarks to the Author:

This study by Cox et al. provides extensive datasets on the pathology of the morbillivirus canine distemper virus (CDV) in ferrets, its immunosuppressive nature and ability to cause immune amnesia, and how these effects can be alleviated therapeutically by a previously developed oral small-molecule inhibitor of the viral polymerase. In addition, they discover exacerbated respiratory disease after consecutive infection with unrelated respiratory viruses (influenza A virus [IAV], respiratory syncytial virus [RSV]) followed by CDV infection.

This study is highly relevant and extremely important to the field for several reasons:

- (1) The authors develop a new ferret infection model using a recombinant CDV with truncated nucleoprotein that overcomes the issue of 100% lethality of wild type CDV infection in ferrets, that thus far has prohibited extensive studies on long-term immunosuppressive effects in this animal model. The new model is improved over existing other models such as nectin-4 blind CDV, which has a restricted cell tropism. The new model for this reason is the closest animal model to measles virus pathology so far except for non-human primate models.
- (2) The authors show that pre-existing humoral immunity against rabies virus and IAV is eradicated by CDV infection and does not recover over prolonged periods (at least 3 months), while CDV infection itself or subsequent infection with other viruses allows generation of novel humoral immune responses.
- (3) The authors establish treatment schedules with the polymerase inhibitor GHP-88309 that not only reduces disease severity, but also prevents immunosuppression and immune amnesia when applied during the lymphatic phase of CDV infection (within 5 days post infection). Previous inhibitors such as ERDRP-0519 were efficacious at earlier treatment starts. Thus, GHP-88309 is a promising antiviral that may have positive effects when given at diagnosis of infection.
- (4) The authors discover that CDV infection causes exacerbated respiratory disease (including hemorrhage) in animals that were previously infected with RSV or IAV. This has not yet been described and provides another piece of evidence for the concerns related to resurgence of measles virus in the human population, although at this point it is unclear whether this phenomenon of CDV can also be observed with measles virus. Importantly, treatment with GHP-88309 also reduces the severity of this complication.

While the overall impact and soundness of this study is not in question, I have a few specific points that confused me:

- (1) In Fig. 1 / Extended Data Fig. 2, the animals that were newly vaccinated against CDV at 28/14 days prior to challenge (orange group) exhibit some weight loss, lymphopenia, and PBMC viremia, while the group that was vaccinated by the breeder (brown group) did not. Do these groups have significantly different anti-CDV antibody titers at the time of challenge, or are there other reasons for these differences?
- (2) Why is the nectin-4-blind CDV in Fig. 2 not inducing lymphocytopenia and immune amnesia? This contradicts earlier studies by Petrova et al. (10.1126/sciimmunol.aay6125) and Sawatsky et al. (10.1128/JVI.06414-11) that used nectin-4-blind CDV in their studies and showed that this virus caused lymphocytopenia like wild type virus and induced immune amnesia. Did the authors use a virus genetically similar or different from the other studies? If the nectin-4-blind mutations are identical, do you have a different explanation for this discrepancy?
- (3) The authors claim that lung pathology in CDV only-infected animals was significantly lower than in those infected consecutively with IAV and CDV (lines 192-197). However, histopathology evaluation in Extended Data Fig. 7b shows data for CDV only on day 19, which is slightly higher than for the IAV/CDV group at the same day, contradicting the above statement, which seems based on comparison of day 15 for the IAV/CDV group with day 19 for the CDV only group. Do you have data for day 15 of the CDV only group for clarification?
- (4) The authors identify reduced IL-1beta expression as a result of IAV pre-infection, while CDV infection upregulates IL-1beta. In addition, upregulation of trefoil factor peptides TFF1, TFF2, and TFF3 after CDV infection is reduced, if animals were previously infected with IAV. Is there a connection of

IL-1b and TFFs? Is upregulation of TFFs dependent on IL-1b, or vice versa? While there is no need to address this experimentally in this study, a few additional sentences in the discussion might be good.

Minor comments:

(5) Lines 124-125: The sentence is confusing, since GHP-88309-treated animals in Fig. 2 do develop clinical signs, only the CDV-vaccinated group did not develop clinical signs.

(6) Lines 130-131: Data in Supplementary Fig. S5a-d does not seem to include groups that were treated with GHP-88309.

(7) Supplementary datasets on RNAseq analysis: I was unable to find any form of legend that would allow to identify from which experimental groups the different datasets were derived. I only saw numbers, which I believe might be animal identifiers.

(8) Line 142: Flagellin data is shown for 1, 2, and 24 h post injection, not 1, 12, and 24h.

(9) Lines 153-155: If data with IAV neutralizing antibody titers after the H1N1 challenge is presented somewhere, I was unable to find it. Fig. 2j only included data until day 63, which is prior to IAV challenge.

(10) Line 188-189: Can you provide some images of necropsies from ferrets infected with CDV only?

(11) Line 190: Type: 10 days should read 12 days

(12) Line 199: Gram-staining (Fig. 3e, Supplementary Fig. 12): What re the yellow and purple colors? Please add explanation to Figure legend.

Response to reviews

Reviewer #1

In this paper Cox et al used canine distemper virus (CDV) infection in ferrets to model measles virus (MeV) infection. The study demonstrates the effectiveness of broad-spectrum paramyxovirus inhibitor GHP-88309 in multiple ways. Firstly, it demonstrates the effectiveness of this drug in reducing CDV lethality. Secondly, it demonstrates its effectiveness in preventing CDV induced immune amnesia and lastly, the drug mitigates exacerbated diseases outcomes from secondary CDV infection following primary influenza infection. The study was very thoroughly done and have explored several valuable questions pertaining to morbillivirus disease. However, the presentation of the data can be a little difficult to follow. Overall, highly valuable work that should be ready for publication with a few tweaks.

There are a few minor comments below that can help improve the clarity of this paper:

a) Clinical scores are recorded in some figures (extended figure 2 etc.) however there is no scoring rubric described in the method. How was severity of rash, conjunctivitis and diarrhea ascertained? Some description of scores would improve the interpretation of this figure.

We have added a scoring rubric for clinical signs (rash, conjunctivitis, diarrhea) to the revised Methods section that is included in the Supplementary Information.

b) Supplementary figure 1d: the last column on right of the bar graph has no label

We apologize for the oversight and have corrected the figure.

c) Figure 1a: the symbols next to the animals in each box presumably shows clinical signs? The vehicle treated animal have no symbols next to it though, even though they presented all symptoms? This is not very clear overall.

We are sorry for the misunderstanding. The graphical study design legend included in figure 1a was meant to define symbol usage, but can be confusing without further explanation. The symbols in the different boxes show clinical presentation of animals when treatment of the respective group was initiated. Since animals in the vehicle and vaccinated groups did not receive drug at any time, no clinical sign symbols are shown for these. We have revised the figure legends for clarification.

d) Line 71, please replace 'Neither' with 'neither'

Corrected.

e) Figure 2e: what do the symbols on top of the timeline mean? The nasal lavage symbol makes sense, but the other shapes are not clearly explained anywhere.

The symbols refer to the different treatment groups, defined in the legend at the bottom of the figure. We have added an explanation to the revised figure legend.

f) Figure 3: This figure and section of the study was least clear. How many groups are there? What does CDV naïve/no flu and CDV naïve/flu mean? Why are there other groups such as CDV vaccinated/flu in extended figure 6 but not in figure 3? Schematic of the groups like in Figure 1 or 2 would have helped with this.

Moreover, in figure 3F, what does the two sections (influenza and CDV disease) mean on top for the CDV only group? Does this mean no prior influenza infection was done? Again, lack of clear description of how many groups there are makes this very confusing to interpret.

We apologize for the lack of clarity in the presentation of these experiments. For this revision of figure 3, we have:

- added a schematic to figure 3a defining the different experimental groups. For the study depicted in figure 3a, there were 3 groups – animals receiving pdm09 H1N1 followed by CDV (flu + CDV), animals receiving CDV only (CDV only), and animals receiving pdm09 H1N1 only (flu only).
- We have standardized our nomenclature throughout the figures and now refer to CDV naïve/no flu as “CDV only”, CDV naïve/flu as “flu + CDV”, and CDV naïve/no CDV as “flu only” throughout.
- In some experiments such as those shown in Extended Data Fig. 6, we added an additional group of reference animals that were CDV vaccinated and received flu followed by CDV, referred to as “CDV vaccinated: flu + CDV”.
- The original labeling of figure 3F was unfortunate indeed. We have removed the confusing labels, now consistently defining the different study periods relative to the day of CDV infection.

g) Figure 3F: The disappearance of Rodentibacter at 0dpi compared to -28 and -10 dpi stands out distinctly but is not commented on. Any thoughts on this?

Samples at 0 dpi were extracted in a terminal BALF procedures from animals euthanized on that day, in contrast to survival BALFs taken at the other time points shown in the figure. Upon reanalysis of the data for this revision, we realized that this difference in procedure affects sample composition, presumably reflecting

that in terminal BALF the entire lung was inflated with PBS, allowing for sampling of the microbial environment deep within the lower respiratory tract, whereas survival BALF predominantly samples the microbial flora present in the trachea and tissue surrounding the bifurcation of the lungs. For a comparison of equivalent samples, we have revised the figure, now showing survival BALF samples only.

h) Supplementary figure S4a: CDV nectin 4 blind nasal titres are missing. Overall comment about S4 is that data from animals infected with WT CDV is missing and should be included.

Since the nectin 4-blind CDV cannot enter epithelial cells from the basolateral side, this recombinant does not shed, making titration from nasal lavages impossible. Animals infected with WT CDV succumb to the infection by 12 dpi, which prevents assessment of most parameters monitored in supplementary figure S4. For this revision, we have, however, added a WT CDV group to panels S4a and b (nasal lavage titer and rash, respectively), which was determined up to death of the animals, and specified in the figure legends which groups are shown in individual subpanels.

i) Line 129, GHP treatment up to 10 dpi reduced shedding in nasal lavages, but it should be noted that titres in PBMCs remain unchanged compared to vehicle only group.

We fully agree, and had therefore commented on unchanged PBMC viremia titers in the 10 dpi-group in the sentence just preceding line 129: "Oral GHP-88309 started 5 dpi significantly reduced CDV viremia titers (Extended Data Fig. 4c), but later initiation of treatment did not affect severity or duration of primary viremia. However, first administration of GHP-88309 up to 10 dpi statistically significantly shortened virus shedding into nasal lavages (Fig. 2g)." We feel this statement addresses the request without further revision.

j) Line 131, Supplementary figure 5a-d deal with CBC in animals infected with different CDV strains and doesn't deal with animals treated with GHP. Needs to be removed from this line and mentioned in relevant earlier sections.

We have corrected the placement of the pointer to Supplementary Fig. S5a-d.

k) Was whole genome sequencing done on only ferret nasal wash samples?

Correct, only genomes of virions shed into nasal lavages were sequenced.

l) Line 142, The 12 hour data of proinflammatory cytokines is not shown in supplementary figure S6.

We apologize for the typo. This line should have read "...harvested before, or 1, 2, and 24 hours after...", and has been corrected in this revision.

m) Line 153, H1N1 antibody titres are said to be present following infection but no data is shown?

Added in this revision as new Extended Data Fig. 5c. The following subpanels of Extended Data Fig. 5 have been relabeled accordingly.

n) Figure 3h: Was TFF levels measured in RSV+CDV combinations to test the hypothesis that reduced TFF levels are linked to exacerbated disease outcomes?

In the chronology of this study, the RNAseq-screen with flu + CDV-infected animals that identified the mechanistic link to TFF expression was carried out after confirmation of the hemorrhagic pneumonia phenotype in the RSV + CDV study. TFF levels in RSV + CDV were therefore not tested at the time, and RNA samples for retroactive assessment are not available.

o) Supplementary figure S13(a-e): Though some differences can be observed in IAV experienced animals, the data is very variable making it difficult to make reliable conclusions regarding the anti-inflammatory state of animals at time of CDV infection. The differences in fold changes are also quite small (4 fold increase in TGF β in IAV animal vs 2-fold in CDV only). This data either needs to be tightened with more repeats or the language needs to be softened away from hard conclusions.

We agree and have toned-down the conclusions drawn from this analysis as recommended. Since the different stages of host respiratory immune responsiveness after resolution of IAV pneumonia have been well documented by others in previous work (for instance, work described in reference 33) and our results matched these data, we decided against committing additional animals to this part of our study.

Reviewer #2

This study by Cox et al. provides extensive datasets on the pathology of the morbillivirus canine distemper virus (CDV) in ferrets, its immunosuppressive nature and ability to cause immune amnesia, and how these effects can be alleviated therapeutically by a previously developed oral small-molecule inhibitor of the viral polymerase. In addition, they discover exacerbated respiratory disease after consecutive infection with unrelated respiratory viruses (influenza A virus [IAV], respiratory syncytial virus [RSV]) followed by CDV infection.

This study is highly relevant and extremely important to the field for several reasons:

- (1) The authors develop a new ferret infection model using a recombinant CDV with truncated nucleoprotein that overcomes the issue of 100% lethality of wild type CDV infection in ferrets, that thus far has prohibited extensive studies on long-term immunosuppressive effects in this animal model. The new model is improved over existing other models such as nectin-4 blind CDV, which has a restricted cell tropism. The new model for this reason is the closest animal model to measles virus pathology so far except for non-human primate models.*
- (2) The authors show that pre-existing humoral immunity against rabies virus and IAV is eradicated by CDV infection and does not recover over prolonged periods (at least 3 months), while CDV infection itself or subsequent infection with other viruses allows generation of novel humoral immune responses.*
- (3) The authors establish treatment schedules with the polymerase inhibitor GHP-88309 that not only reduces disease severity, but also prevents immunosuppression and immune amnesia when applied during the lymphatic phase of CDV infection (within 5 days post infection). Previous inhibitors such as ERDRP-0519 were efficacious at earlier treatment starts. Thus, GHP-88309 is a promising antiviral that may have positive effects when given at diagnosis of infection.*
- (4) The authors discover that CDV infection causes exacerbated respiratory disease (including hemorrhage) in animals that were previously infected with RSV or IAV. This has not yet been described and provides another piece of evidence for the concerns related to resurgence of measles virus in the human population, although at this point it is unclear whether this phenomenon of CDV can also be observed with measles virus. Importantly, treatment with GHP-88309 also reduces the severity of this complication.*

While the overall impact and soundness of this study is not in question, I have a few specific points that confused me:

- (1) In Fig. 1 / Extended Data Fig. 2, the animals that were newly vaccinated against CDV at 28/14 days prior to challenge (orange group) exhibit some weight loss, lymphopenia, and PBMC viremia, while the group that was vaccinated by the breeder (brown group) did not. Do these groups have significantly different anti-CDV antibody titers at the time of challenge, or are there other reasons for these differences?*

The vendor (Triple F farms) uses the Distemink CDV vaccine, which could not be sourced by us. We therefore used Merial's Purevax Ferret Distemper vaccine. Both vaccines are approved for veterinary use against CDV infection. We have clarified the different vaccine types in the revised text, and furthermore added an additional subpanel to revised Supplementary Fig.3, confirming that at study start (time of CDV infection) there were no significant differences in nAb levels between animals vaccinated by the vendor and those vaccinated by us 28 and 14 days before CDV infection.

- (2) Why is the nectin-4-blind CDV in Fig. 2 not inducing lymphocytopenia and immune amnesia? This contradicts earlier studies by Petrova et al. (10.1126/sciimmunol.aay6125) and Sawatsky et al. (10.1128/JVI.06414-11) that used nectin-4-blind CDV in their studies and showed that this virus caused lymphocytopenia like wild type virus and induced immune amnesia. Did the authors use a virus genetically similar or different from the other studies? If the nectin-4-blind mutations are identical, do you have a different explanation for this discrepancy?*

Genetically identical viruses (recCDV-5804p background) and Nectin-4 blinding mutations were used in the studies referenced by the reviewer and in our work. Close examination of the published work and our data reveals that no actual discrepancy in lymphocytopenia exists between the different studies. Specifically, figure 6D of PMID: 31672862 (Petrova et al, Science Immunology, 2019) shows an approximately 1 order of magnitude reduction in circulating PBMCs after infection of ferrets with recCDV Nectin 4-blind. In figure 2c, we show an equal (approximately 1 order of magnitude) reduction after infection with recCDV Nectin 4-blind virus. The notion of a seeming discrepancy in results may have arisen from the fact that Petrova et al plotted their data on a linear-scale y-axis whereas we present our results on a log-scale y-axis, which was better suited to show the, in comparison, substantially greater reduction in PBMCs after infection with recCDV-5804p WT and recCDV-5804p N Δ 425-479 (over 2 orders of magnitude). Consistent with much more severe lymphocytopenia after infection of ferrets with recCDV-5804p N Δ 425-479 compared to recCDV Nectin 4-blind, immune amnesia was more pronounced and permanent rather than transient in these animals (Fig. 2d).

- (3) The authors claim that lung pathology in CDV only-infected animals was significantly lower than in those infected consecutively with IAV and CDV (lines 192-197). However, histopathology evaluation in Extended Data Fig. 7b shows data for CDV only on day 19, which is slightly higher than for the IAV/CDV group at the same day, contradicting the above statement, which seems based on comparison of day 15 for the IAV/CDV group with day 19 for the CDV only group. Do you have data for day 15 of the CDV only group for clarification?*

We euthanized flu + CDV-infected animals on 5 dpi with CDV or when they became moribund and reached predefined endpoints. In this particular study, individual animals became moribund 15, 18, and 19 dpi. When the last animal in the flu + CDV-infected group reached endpoint 19 dpi, we also sacrificed all animals of the CDV only, CDV vaccinated, GHP-88309-treated groups. We have revised the Extended Data Fig. 7a, now clarifying the timeline breakdown for individual animals.

- (4) The authors identify reduced IL-1beta expression as a result of IAV pre-infection, while CDV infection upregulates IL-1beta. In addition, upregulation of trefoil factor peptides TFF1, TFF2, and TFF3 after CDV infection is reduced, if animals were previously infected with IAV. Is there a connection of IL-1b and TFFs? Is upregulation of TFFs dependent on IL-1b, or vice versa? While there is no need to address this experimentally in this study, a few additional sentences in the discussion might be good.*

We had summarized in the Discussion that reportedly type 2 cytokines IL-4 and IL-13 stimulate TFF3 synthesis in the GI tract, but very little is known about the regulation of TFF expression in the respiratory tract in any species. We have added the possibility of a connection between IL-1 β levels and TFF upregulation to the revised Discussion as recommended, but developing a mechanistic understanding of TFF expression in the respiratory tract will ultimately be the objective of future work.

Minor comments:

(5) Lines 124-125: *The sentence is confusing, since GHP-88309-treated animals in Fig. 2 do develop clinical signs, only the CDV-vaccinated group did not develop clinical signs.*

We agree. The statement “without developing clinical signs” was meant to apply to the CDV-vaccinated group only, but has been deleted in this revision to avoid misunderstandings.

(6) Lines 130-131: *Data in Supplementary Fig. S5a-d does not seem to include groups that were treated with GHP-88309.*

We apologize for the misplaced pointer to Supplementary Fig. S5A-d. As explained in our response to comment (j) from reviewer #1, we have corrected the placement of the figure pointer in this revision. Results for GHP-88309-treated groups are shown in Fig. 2 and Extended Data Fig. 4.

(7) *Supplementary datasets on RNAseq analysis: I was unable to find any form of legend that would allow to identify from which experimental groups the different datasets were derived. I only saw numbers, which I believe might be animal identifiers.*

We have added a legend for Supplementary Datasets 1 and 3 to the revised Supplementary Information. Specifically, numbers specify individual animals, D# refers to the day of sample extraction after CDV infection.

- Legend for Supplementary Dataset 1:
425, input virus for infection; 626.D8, 628.D8, RNA isolated 8 dpi from ferrets first treated with GHP-88309 5 dpi; 458.D8, 638.D8, 639.D8, RNA isolated 8 dpi from ferrets first treated with GHP-88309 7 dpi; 640.D9, 641.D9, 642.D9, RNA isolated 9 dpi from ferrets first treated with GHP-88309 24 dpi.
- Legend for Supplementary Dataset 3:
425, input; 538.Dx, 539.Dx, 540.Dx, RNA isolated x dpi from ferrets first treated with GHP-88309 7 dpi; 542.Dx, 543.Dx, 544.Dx, RNA isolated x dpi from ferrets first treated with GHP-88309 10 dpi; 545.D11, 546.D11, 547.D11, RNA isolated 11 dpi from ferrets that were vehicle treated; 548.D11, 549.D11, 550.D11, RNA isolated 11 dpi from ferrets first treated with GHP-88309 14 dpi; 550.D19, RNA isolated 19 dpi from ferrets first treated with GHP-88309 14 dpi.

(8) Line 142: *Flagellin data is shown for 1, 2, and 24 h post injection, not 1, 12, and 24h.*

We apologize for the typo. Corrected in this revision.

(9) Lines 153-155: *If data with IAV neutralizing antibody titers after the H1N1 challenge is presented somewhere, I was unable to find it. Fig. 2j only included data until day 63, which is prior to IAV challenge.*

We apologize for the confusion. As explained in our response to comment (m) of reviewer #1, we have added these IAV nAb titers in this revision as new Extended Data Fig. 5c. The following subpanels of Extended Data Fig. 5 have been relabeled accordingly.

(10) Line 188-189: *Can you provide some images of necropsies from ferrets infected with CDV only?*

We employed non-invasive pulmonary ferret MRI to reduce the number of animals required for this study. Accordingly, we do not have necropsies of animals infected with CDV only for the 10-18 dpi time window, since these animals did not become moribund. Lung tissues of animals infected with CDV only that were extracted 19 dpi were macroscopically unremarkable.

(11) Line 190: *Type: 10 days should read 12 days.*

Corrected in this revision.

(12) Line 199: *Gram-staining (Fig. 3e, Supplementary Fig. 12): What are the yellow and purple colors? Please add explanation to Figure legend.*

Stain colors are explained in the revised legend to Fig. 3e.

Reviewers' Comments:

Reviewer #1:

Remarks to the Author:

Most of the concerns regarding clarity has been addressed and the article is suitable for publication.

Reviewer #2:

Remarks to the Author:

Thank you for addressing all comments!